



**Greenhouse gas emissions and their trends over the last three decades across**
**Africa**
Mounia Mostefaoui[1], Philippe Ciais[2], Matthew J. McGrath[2], Philippe Peylin[2], Prabir K.
Patra[3], Yolandi Ernst[4]
[1]Laboratoire de Météorologie Dynamique/IPSL, École Normale Supérieure, PSL Research University,
Sorbonne University, École Polytechnique, IP Paris, CNRS, Paris, France.
[2]Laboratoire des Sciences du Climat et de l'Environnement, 91190 Gif-sur-Yvette, France.
[3]Research Institute for Global Change, JAMSTEC, Yokohama 2360001, Japan.
[4]Global Change Institute, University of the Witwatersrand, Johannesburg, South Africa.
*Correspondence to*: mounia.mostefaoui@polytechnique.edu
**Key words:** greenhouse gasses, anthropogenic emissions and removals, fossil fuels, land-use, land-use change and
forestry, Africa, bottom-up, top-down atmospheric inversions, UNFCCC inventories, Global Carbon Project,
PRIMAP-hist, IPCC sectors, climate change, Paris Agreement, Global Stocktake, Monitoring, Reporting and
Verification.
**Abstract.** A key goal of the Paris Agreement (PA) is to reach net-zero Greenhouse Gasses (GHG) emissions by 2050
globally, which requires mitigation efforts from all countries. Africa's rapidly growing population and GDP makes
this continent important for GHG emission trends. In this paper, we study the emissions of carbon dioxide ($CO_2$),
methane ($CH_4$) and nitrous oxide ($N_2O$) in Africa over three decades (1990-2018). We compare bottom-up approaches
including UNFCCC national inventories, FAO, PRIMAP-hist, process-based ecosystem models for $CO_2$ fluxes in the
Land Use, Land Use Change and Forestry (LULUCF) sector, and global atmospheric inversions. Our database is
available from Zenodo at: https://doi.org/10.5281/zenodo.7347077 (Mostefaoui et al., 2022). For inversions, we
applied different methods to separate anthropogenic $CH_4$ emissions. The bottom-up (BU) inventories show that over
the decade 2010-2018, less than ten countries represented more than 75% of African fossil $CO_2$ emissions. With a
mean of 1373 $MtCO_2$ $yr^{-1}$, total African fossil $CO_2$ emissions over 2010-2018 represent only 4% of global fossil
emissions. Yet, these emissions grew by +34% from 1990-1999 to 2000-2009 and by +31% over 2000-2009 to 2010-
2018, more than doubling in 30 years. This growth rate is more than twice faster than the global growth rate of fossil
$CO_2$ emissions. The anthropogenic emissions of $CH_4$ grew by 5% from 1990-1999 to 2000-2009 and by 14.8% from
2000-2009 to 2010-2018. The $N_2O$ emissions grew by 19.5% from 1990-1999 to 2000-2009; and by 20.8% from
2000-2009 to 2010-2018. When using the mean of estimates from UNFCCC reports (including the land use sector),
with corrections from outliers, Africa was a mean source of greenhouse gasses of $+2622^{3239}_{2186}$ $MtCO_2e$ $yr^{-1}$ from all
bottom-up estimates (sub- and superscript indicating min-max range uncertainties), and of $+2637^{5873}_{1761}$ $MtCO_2e$ $yr^{-1}$
from top-down methods, during their overlap period from 2001 to 2017. Although the mean values are consistent, the
range of top-down estimates is larger than the one of bottom up, indicating that sparse atmospheric observations and
transport model errors do not allow us to use inversions to reduce the uncertainty of bottom-up estimates. A main
source of uncertainty comes from $CO_2$ fluxes in the land-use sector (LULUCF) for which the spread across inversions
is larger than 50%, especially in Central Africa. Moreover, estimates from national UNFCCC communications differ



widely depending on whether the large sinks in a few countries are corrected to more plausible values using more
recent national sources following the methodology of Grassi et al. (2022) The median of $CH_4$ emissions from
inversions based on satellite retrievals and in situ surface networks are consistent with each other within 2% at
continental scale. The inversion ensemble also provides consistent estimates of anthropogenic $CH_4$ emissions with
bottom-up inventories such as PRIMAP-hist. For $N_2O$, inversions systematically show higher emissions than
inventories, on average about 4.5 times more than PRIMAP-hist, either because natural $N_2O$ sources cannot be
separated accurately from anthropogenic ones in inversions, or because bottom-up estimates ignore indirect emissions
and under-estimate emission factors. Future improvements can be expected thanks to a denser network for monitoring
atmospheric concentrations. This study helps to introduce methods to enhance the scope of use of various published
datasets and allows to compute budgets thanks to recombinations those data products. Our results allow to understand
uncertainty and trends of emissions and removals in a region of the world where few observations exist and most
inventories are based on default IPCC guidelines values. The results can therefore serve as a support tool for the Global
Stocktake (GST) of the Paris Agreement. The referenced datasets related to figures are available at:
https://doi.org/10.5281/zenodo.7347077 (Mostefaoui et al., 2022).



**Introduction**

Large global reductions of greenhouse gasses (GHG) emissions are needed to avoid "dangerous anthropogenic interference with the climate system" (IPCC, 2021). The Paris Agreement (PA) aims at limiting global warming below 2°C and reaching "net-zero GHG emissions by 2050" (UNFCCC, 2015). To improve the monitoring of emissions trends, the PA has an Enhanced Transparency Framework (ETF) by which countries will have to report their GHG emissions and removals under a standardized format starting in 2024 (Perugini et al., 2021; UNFCCC, 2021) through Biennial Transparency Reports (BTR), with the ambition to use up-to-date data and best available science to improve national inventories. This represents a challenge for many developing countries, where emissions inventories have been irregular.

Recent analyses predict a fast increase of African emissions correlated with demographic growth. The African population is expected to double from 1.2 billion in 2019 to 2.5 billion at the 2050 horizon (UN, 2019). Using the TIAM-ECN Integrated Assessment Model (IAM) developed with data from the International Energy Agency (IEA), van der Zwaan et al., (2018) concluded that greenhouse gasses (GHG) emissions from Africa will become substantial at the global scale by 2050. In Shared Socio-economic Pathways (SSP) projection scenarios, Africa and the Middle East are grouped together despite having very different geographies, per capita emissions and Gross Domestic Product (GDP) (IIASA, 2017). According to IAM projections, the minimum projected share of Africa in global emissions would be close to 10% by 2050 for a business-as-usual pathway. An "explosive growth in African combustion emissions'" (Liousse et al., 2014) could not be excluded from 2030 to 2050, if no drastic mitigation policies are implemented (IPCC, 2021). If a stringent emissions reduction pathway limiting global warming to +2 °C is adopted, Africa could contribute to around 20% of global emissions by 2050, becoming the second largest worldwide emitting region. Further, under stringent climate policy scenarios, $CH_4$ and $N_2O$ emissions in Africa were projected to contribute 80% of the total emissions of these two gasses in 2050 (van der Zwaan et al., 2018). Therefore, Africa will become an important global emission contributor under any mitigation pathway with a demographical and industrial development increase.

There are 56 African countries represented in the United Nations. National emissions reports to the United Nations Convention Framework on Climate Change (UNFCCC) are available for 53 countries, including all major African emitters. Africa as a whole ranks fifth worldwide in terms of territorial fossil fuels use with a total of 1449 $MtCO_2e$, in-between the Russian Federation and Japan (Friedlingstein et al., 2020). The global share of Africa is ∼4% of fossil $CO_2$ ($FCO_2$) emissions, ∼16





% of $CH_4$ emissions (Saunois et al., 2020) and ∼25% of $N_2O$ emissions (Tian, 2020). South Africa is
the biggest $FCO_2$ emitter in the continent, and ranked twelve on the global scale, just after Brazil.
Despite projections of strong growth of emissions and population in Africa, the continent is under-
studied and lacks up-to-date comprehensive assessments of GHG emissions and removals, given
sporadic and often outdated reports by individual countries. The literature tends to be scarce about
African countries, and their emissions have rarely been analyzed comprehensively using the results
from both statistical inventories that are also referred to as bottom-up (BU) methods, and from top-
down (TD) atmospheric inversions. Inversions results are uncertain due to the small number of
atmospheric stations over the continent (Nickless et al., 2020). A previous analysis of African emissions
was solely focused on $FCO_2$ emissions during the decade 2000-2009 (Canadell et al., 2009). A first
budget for the period 1990-2009 was provided at the continental scale with the RECCAP1 project
(Valentini et al., 2014). Ayompe et al. (2020) studied recent $FCO_2$ emissions trends, using International
Energy Agency (IEA) data. Other studies are region-specific or sector-specific, focusing exclusively
on agriculture (Bombelli et al., 2009), on natural ecosystems in Sub-Saharan Africa (Kim et al., 2016)
or in individual countries such as Kenya (Zhu et al., 2018).
Paying attention not only to commonly identified big emitters like South Africa, but also to medium
emitters and to emerging emitters is important, not only in terms of scientific assessment, but also for
financial and climate policy purposes under the PA. The Monitoring, Reporting and Verification
(MRV) provisions of the PA indeed require scientific and policy tools to verify the pledges made by
all the signatory countries. Instruments for financial transfers for mitigation and adaptation like the
Green Fund on Climate Change (GCF) and the REDD+ initiatives cover the African scope and will
require scientific assessment of trends for impact evaluation and credibility purposes, and as an
incentive for continued investments. As part of the Global Stock Take (GST) under article 14 of the
PA aiming at assessing "collective progress", all signatory parties will have to show their contributions
to the global mitigation efforts. These efforts will be evaluated within a MRV system which includes
the requirement for developing countries to submit their Biennial Update Reports (BUR) on a biennial
basis starting in 2024. As no standard global reporting framework has been required to date, we
anticipate that the data available for the first stocktake in 2023 will be very heterogeneous. As a
continent gathering non-Annex I countries exclusively, the African case is featured by the scarcity of
national official inventories which have been provided to date on a voluntary basis through National
Communication (NC) and BUR. BU estimates of emissions established by independent scientific
methods are also discussed in the present study. In this context, different and complementary
observation-based methods assessing national GHG emissions and sinks are needed.



The aim of this paper is to evaluate relative merits of different existing types of datasets for the
assessment of African emissions and removals and their trends for $CO_2$, $CH_4$ and $N_2O$ during the last
three decades. In this paper, we standardize the metrics and scope of application for different categories
of GHG emissions to discuss budgets. We also validate and benchmark different independent datasets
to evaluate the possibility to use them as a verifying tool for official country-reported data. In order to
cover all GHG sectors, we also describe recombinations of different historical datasets for the last 30
years that are necessary to fill the gap for some missing past sectorial emissions. This study offers a
comparison of data products originally combined to compute a budget and an evaluation of their
relative merits. The different data products discussed here include different bottom-up (BU)
approaches, including official countries communications to the UNFCCC and estimations from the
Food and Agriculture Organization (FAO), Carbon Dioxide Information Analysis Center (CDIAC),
global inventories for anthropogenic emissions (PRIMAP-hist which integrates combinations of
various datasets including FAO and Global Carbon Project (GCP)), and process-based models for land
$CO_2$ fluxes with 14 Dynamic General Vegetation Models (DGVM) from the TRENDY version 9
ensemble (Table 1). We also analyze and combine top-down data products to discuss individual gas
and to compute budgets: three atmospheric global inversions for $CO_2$ land fluxes; 22 inversions for
$CH_4$ emissions (11 in situ inversion models and 11 satellite inversion models) and $CH_4$ wildfire
emissions from the Global Fires Emission Dataset (GFED) version 4. We used three inversion models
for $N_2O$ fluxes (PyVAR model, TOMCAT-INVICAT model, and MIROC4-ACTM model (see Table
1). Inversions only solve for total fluxes or at best for groups of sectors, whereas BU estimates have a
larger number of sectors. In Table 2, we present the correspondence between 'sectors' defined by the
TD and BU methods. For all datasets, we chose an atmospheric convention with negative values
representing removals from the atmosphere (i.e. land sink). We deliver and original comparison of BU
estimates from national inventories, global inventories, and process-based models, with TD estimates
from atmospheric inversions over Africa. The work is carried out for large countries or groups of small
countries, as inversions do not have the capability to constrain fluxes over small areas given their coarse
grid and sparse atmospheric data. Based on the benchmarking and relative merits evaluation of the
various data products presented above, the scientific questions addressed in this study are: 1) How
consistent are the mean values and trends of GHG emissions across BU estimates in Africa? 2) How
consistent are the different inversion model results? 3) How do inversions compare with bottom-up
estimates? 4) What is the net GHG balance of the African continent from different observation-based
methods, including $CO_2$ sinks and sources in the land-use sector? 5) What are the main sources of
uncertainties?



The manuscript is organized into two main sections. First, a material and methods section describes the regional breakdown and input data (section 1). We present our results for the whole Africa and for six groups of aggregated countries (section 2) with a specific analysis of $CO_2$ emissions and sinks, divided between $FCO_2$ (section 2.1), fluxes in the land use, land use change and forestry (LULUCF) sector (section 2.2), and emissions of non-$CO_2$ greenhouse gasses (sections 2.3 and 2.4). Conclusions are drawn about uncertainties of African GHG net emissions and removals assessment.

## 1 Methods and datasets

This study covers the period from 1990 to 2018, and emissions and sinks of $CO_2$, $CH_4$ and $N_2O$. We used 1990 as a base year since reporting to the UNFCCC mostly started in that year and is often used as a reference comparison year in national pledges of the PA. The last year of analysis is 2018, reflecting the availability of inversion data and avoiding further uncertainty due to poorly understood emissions changes before and after the COVID19 crisis. This period allows the analysis of decadal features. It also has the advantage of being covered by several datasets, listed in Table 1. We considered different bottom-up (BU) approaches, including official countries communications to the UNFCCC and estimations from the Food and Agriculture Organization (FAO), global inventories for anthropogenic emissions (PRIMAP-hist which integrates combinations of various datasets including FAO, GCP, EDGAR v4.3.2, Andrew 2018 cement data, Biennal Updtaed Reports (BUR), Common Reporting Format (CRF), UNFCCC data, and BP), and process-based models for land $CO_2$ fluxes with 14 Dynamic General Vegetation Models (DGVM) from the TRENDY version 9 ensemble (Table 1). We used three atmospheric global inversions for $CO_2$ land fluxes; 22 inversions for $CH_4$ emissions; and three inversions for $N_2O$ fluxes (Table 1). Inversions only solve for total fluxes or at best for groups of sectors, whereas BU estimates have a larger number of sectors. In Table 2, we present the correspondence between 'sectors' defined by the TD and BU methods. For all datasets, we chose an atmospheric convention with negative values representing removals from the atmosphere (i.e. land sink). No specific standard guidelines currently exist for defining uncertainties for datasets from BU and TD data products. In general, uncertainty estimates are understood as the spread among minimum and maximum values from one methodology. A main source of uncertainty in the comparison of country-reported data with other data products is the inclusion or not of natural fluxes additionally to anthropogenic emissions sectors. For inversions, the prior geospatial distribution of emissions is a critical source of uncertainty. For the comparability of the different data products presented in this study, we discuss only the mean value over the period of overlapping data availability. Referenced datasets are available at https://doi.org/10.5281/zenodo.7347077(Mostefaoui et al., 2022).





**Table 1. List of BU and TD methods used. (For more details, see also Saunois et al. (2020) for CH₄, Friedlingstein et al. (2020) for FCO₂; UNFCCC country-reported data; Gütschow et al. (2021) for PRIMAP-hist).**

| Dataset name | Method | CO$_2$ | CH$_4$ | N$_2$O | Spatial resolution (longitude × latitude) | Time period covered in the present work |
|---|---|---|---|---|---|---|
| **Inversions** | | | | | | |
| **Global Carbon Budget ensemble (2020)** | TD | × | | | from 1° × 1° to 6°× 4° | 2000-2019 |
| **Global Methane Budget ensemble [1] (2020)** | TD | | × | | from 1° × 1° to 6° × 4° | 2000-2017[2] |
| **PyVAR** | TD | | | × | 3.75° × 1.875° | 1998-2017 |
| **TOMCAT-INVICAT** | TD | | | × | 5.6° × 5.6° | 1998-2015 |
| **MIROC4 -ACTM** | TD | | | × | 2.8° × 2.8° | 1998-2016 |
| **DGVMs** | | | | | | |
| **TRENDYv9 [3]** | BU | | | | 0.5°× 0.5° (land surface) or 1° × 1° | 1990-2019 |
| **Other BU inventories** | | | | | | |
| **PRIMAP-hist (excluding LULUCF)** | BU | × | × | × | country | 1990-2019 |
| **GCB (CDIAC) (excluding LULUCF)** | BU | × | | | 0.1°× 0.1° | 1990-2019 |
| **UNFCCC** | BU | × | | | country | 1990-2015 |
| **FAO (LULUCF CO₂)** | BU | × | | | country | 1990-2019 |
| **GFEDv4 (wildfires only)** | BU | | × | | 0.25°× 0.25° | 1997–2016 |

[1] See 22 inversions details in the supplementary Table S6.

[2] Variations from 2003-2015, 2000-2015, 2010-2017: see detailed period coverage for each dataset in the supplementary Table S6.

[3] See supplementary Table S5 for the 14 products



**Table 2. Sectoral reconciliation between categories defined in TD and BU methods.**

| Gas | Sector label choice for BU and TD | TD inversions | BU inventories |
|---|---|---|---|
| CO₂ | Net land flux | Total Net Biome Productivity (NBP) after subtraction of prior prescribed Fossil CO₂ | Energy + Industrial Processes and Product Use + Agriculture + Waste + Biomass burning |
| CH₄ | Total anthropogenic emissions | Fossil + Anthropogenic Biomass burning (BBUR) + Agriculture & Waste -Wildfires | Energy + Industrial Processes + Agriculture +Waste + Biomass burning |
| N₂O | Total | Total | All IPCC sectors |

**1.1 Regional breakdown**
As some countries are small emitters and their area is too small to be resolved by inversions, and in some
cases even by DGVMs, we grouped African countries into six regions shown in Figure S1 and listed in
Table S1. The grouping followed national borders and biomes similarity considering the Köppen-Geiger
climate zones (Beck et al., 2018), magnitudes of fossil fuel emissions, and per capita emissions (Fig. S1,
Fig. S2 and Fig. S7). We also grouped a maximum of about ten countries per region.
**1.2 Inventories**
**PRIMAP-hist anthropogenic emissions assessment for CO₂, CH₄, and N₂O**
The PRIMAP-hist version 2.2 BU dataset is derived from Gütschow et al., (2021) and combines UNFCCC
reports with a gap-filling method to produce a time series of annual anthropogenic emissions for different
IPCC sectors. PRIMAP-hist does not cover the LULUCF sector for CO₂ due to the high uncertainties.
PRIMAP-hist does not include emissions from shipping and international aviation, but includes cement as
part of FCO₂ emissions. We use data from the HISTCR scenario (data accessed from https://www.pik-
potsdam.de/paris-reality-check/primap-hist/ in April 2022) from country-prioritized dataset, which mainly
uses UNFCCC (BUR and NC) data, unless such data are missing, in that case PRIMAP-hist uses
extrapolated data from EDGAR (2021), FAO (2021) and BP Statistical Review of World Energy (2021).



### Global Carbon Project (GCP) fossil $CO_2$ emissions

We used country-level $FCO_2$ data published by the global $CO_2$ budget by the Global Carbon Project (GCP) (Friedlingstein et al., 2020) separated per fuel type (gas, oil and coal) and including fossil fuel use in the combined industry, ground transportation and power sectors, natural gas flaring, cement production, and process-related emissions (e.g. fertilizers and chemicals). Data for African countries coming among others from the Carbon Dioxide Information Analysis Center (CDIAC) compiled until 2018 (Gilfillan & Marland, 2021), BP Statistical Review of World Energy (BP, 2020), and recent estimates of cement production and clinker-to-cement ratios (Andrew, 2020).

### UNFCCC inventories for $CO_2$ in the LULUCF sector

We used UNFCCC submissions for LULUCF $CO_2$ fluxes from NC and BUR reports downloaded from the UNFCCC website (https://unfccc.int/) in March 2021, and further processed into .csv tables by Deng et al., (2021). Those estimates are based on different accounting methods following the IPCC Guidelines (IPCC, 2006; IPCC, 2019). African countries, being Non-Annex I countries, do not report emissions every year. Figure 1 shows the number of BUR and NC provided each year per African region. The years 1990, 1994, 1995, 2000 and 2005 are featured with several updates, while most of the other years have few updates. About every two years, all regions have at least one update. Note that flexibility for BUR is given to Least Developed Countries (LDCs), that include 33 out of 56 African countries, and to Small Islands Developing States (SIDS), that include six African countries (Table S3).

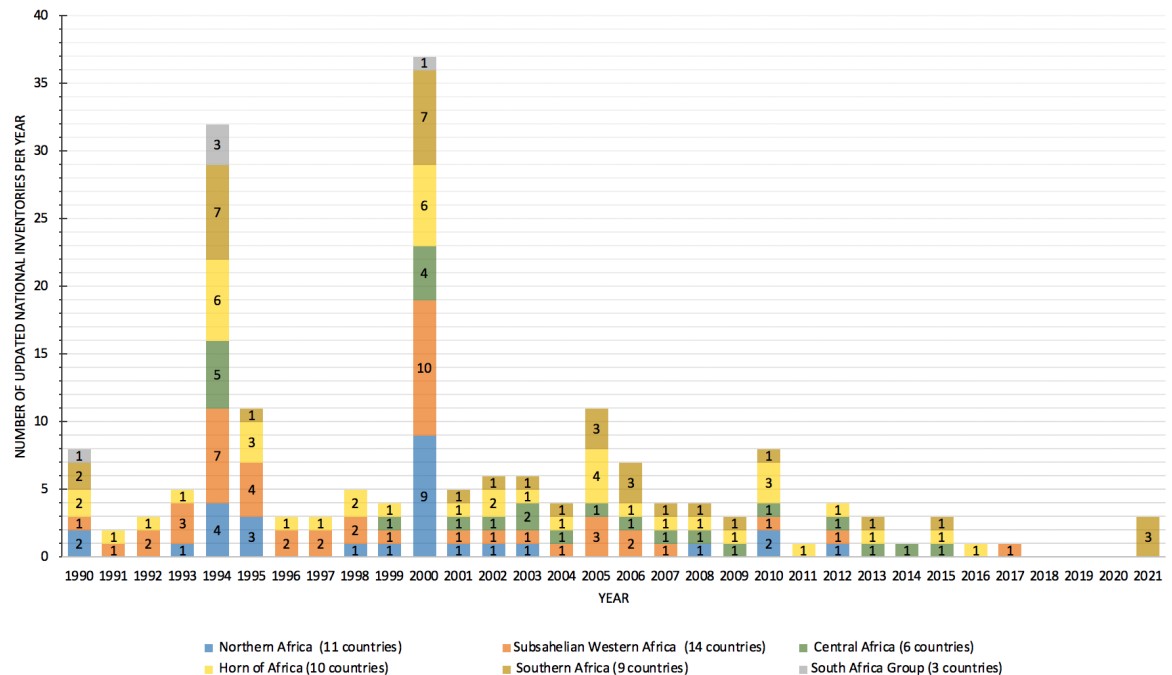

**Figure 1. Number of UNFCCC reports for LULUCF CO$_2$ fluxes in National Communications and Biennial Update Reports, per group of countries defined in Table S1.**

Non-Annex I African countries can use older versions of the IPCC guidelines (IPCC, 2006; IPCC, 2019a). This induces uncertainties from changes in accounting methods between versions, with recent guidelines having more detailed sectors and sources. There is no data for Libya, Equatorial Guinea, Malawi and Sierra Leone during the whole period. UNFCCC data are missing in some years for Rwanda, Sao Tome & Principe, Senegal, South Sudan, Angola. There is no data during 1990-1998 for Liberia.

We noticed that NC and BUR lack details regarding the methods used, the sources for activity data and emissions factors, and most of them are in French language. BUR in .pdf format include a non-standardized table for emissions. The reader is sometimes referred to the "national coordinator for climate change service" with no link to any database or contact person.

Because the PA targets human-induced emissions, countries use the proxy of "Managed lands" for the LULUCF sector, as defined by the IPCC guidelines (https://www.ipcc-nggip.iges.or.jp/public/2006gl/vol4.html; last accessed in August 2022). Managed lands are areas where LULUCF CO$_2$ fluxes are assigned to some anthropogenic activities. Several African NC and BUR do not contain information on their managed lands areas. We thus looked at REDD+ national reports (https://redd.unfccc.int/submissions.html?topic=6; last accessed in August 2022) to get this information (Fig. S2 and Table S8). LULUCF CO$_2$ fluxes on managed lands result from either direct anthropogenic



effect such as land use change and forestry, or indirect effects (such as change in $CO_2$ and climate) on land
remaining in the same land use, e.g. forest remaining forest (Grassi et al., 2022). The vast majority of African
countries use a Tier 1 IPCC accounting method which does not distinguish between these different effects.
Tier 1 methods use a classification with only three out of six possible types of land: "forest land", "cropland"
and "grassland", and do not give spatially explicit land use data. Tier 2 methods include fluxes from six
land use types: forest, cropland and grassland, wetlands, urban and other land-use, for the case of land
remaining under the same land use type, and for the case of conversions between land use types. In Africa,
only South Africa and Zambia used Tier 2 methods for some LULUCF $CO_2$ subsectors.
**Processing of the UNFCCC LULUCF $CO_2$ data and outliers correction**
We processed the UNFCCC LULUCF $CO_2$ data for outlier corrections (Table S4). For Guinea-Bissau, and
Tanzania, we identified inconsistent values from successive communications with substantially differing
numbers. For Guinea, Madagascar, Zimbabwe, Congo, Mali, the Central African Republic (CAF), Angola
and Mauritius we identified changes of more than one order of magnitude between two consecutive reports
and likely implausibly large carbon sinks considering their national forest area. The computations of per
area emissions and removals showed discrepancies, which points out the need for further examination and
inspection of more recent reports in NDC and REDD+ reports (Table S4). Our corrections explained in the
supplementary section are consistent with those proposed by Grassi et al. (2022) who diagnosed
'biophysically impossible' sequestration rates with a threshold value larger than 10 tCO$_2$/ha yr$^{-1}$ over an
area greater than 1 Mha. For Namibia, Nigeria and the Democratic Republic of the Congo (DRC), it was
challenging to select a best estimate between recent and past reports. For those countries, corrections using
more recent data than BUR /NCs have high uncertainties, as noted by Grassi et al. (2022). This includes the
absence of any sink for DRC for instance, contrary to sinks consistently reported over time and large forested
area in this country's previous reports to the UNFCCC. We therefore systematically looked at corrected
values for both case scenarios (with and without Namibia, Nigeria and DRC data corrections). In total, we
corrected 13 outliers as shown in Table S4, consistently with Grassi et al. (2022).
**Food and Agriculture Organization of the United Nations (FAO) LULUCF $CO_2$ fluxes**
We used data from LULUCF $CO_2$ fluxes over 1990-2019 from the FAO Global Forests Resource
Assessments (FAO FRA; data License: CC BY-NC-SA 3.0 IGO, extracted from: https://fra-data.fao.org;
date of Access: May 2022). According to the 2005 FAO categories and definitions, forest is land covering
at least 0.5 hectares and having vegetation taller than 5 meters with a canopy cover higher than 10%. Other
wooded lands refer to land that are not classified as "forest" but that are wider than 0.5 ha, have a canopy
cover of 5%-10% or combine trees, shrubs and bushes with cover higher than 10%. The FAO data for forests



comprise carbon stock changes from both aboveground and belowground living biomass pools. They are
independent from country-reported UNFCCC emissions and removals. The FAO estimates are based on
activity data, areas of forest land and $CO_2$ emissions and removals factors. The FAO data reports: 1) net
emissions and removals from "forest land remaining forest land" and from "land converted to forest"
grouped together, and 2) emissions from "net forest conversion", i.e. deforestation. In contrast, the UNFCCC
accounting uses a 20-years window for $CO_2$ fluxes from land use change, while land-use change fluxes from
land-converted-to-forest are reported separately from those of 'forest remaining forest'.

**1.3 Dynamic Global Vegetation Models (DGVM)**

We used Net Biome Productivity (NBP) from 14 Dynamic Global Vegetation Models (DGVM) from the
TRENDY v9 ensemble covering the period 1990-2019. The different models described in Friedlingstein et
al. (2019) are: CABLE, CLASS, CLM5, DLEM, ISAM, JSBACH, JULES, LPJ, LPX, OCN, ORCHIDEE-
CNP, ORCHIDEE-SDGVM, and SURFEX (Table S5). DGVM are forced by historical reconstructions of
land cover change, atmospheric $CO_2$ concentration and climate since 1901. Detailed cropland management
practices are generally ignored, except for the harvest of crop biomass. Forest harvest is prescribed from
historical statistics in 11 models (Table A1, of Friedlingstein et al., (2020)). The models simulate carbon
stock changes in biomass, litter and soil pools. From the difference between simulations with and without
historical land cover change, a flux called 'land use emissions' can be obtained from DGVM. This flux
includes the indirect effects of climate and $CO_2$ on lands affected by land use change, and a foregone sink
called "loss or gain of atmospheric sink capacity", which is absent from the methods used by UNFCCC and
FAO. Thus, land use change fluxes from DGVM were not compared with other estimates. Note that DGVM
do not explicitly separate managed and unmanaged land. Thus, we used all forest lands to calculate their
mean $CO_2$ fluxes.

**1.4 Atmospheric inversions**

**$CO_2$ inversions**

We used the net land $CO_2$ fluxes excluding fossil fuel emissions (hereafter, net ecosystem exchange) from
three global inversions of the Global Carbon Project that cover a long period (see Table A4 of Friedlingstein
et al., 2020), including : CarbonTrackerEurope (CTRACKER-EU-v2019; van der Laan-Luijkx et al., 2017),
the Copernicus Atmosphere Monitoring Service (CAMSV18-2-2019; Chevallier et al., 2005), and one
variant of Jena CarboScope (JENA, sEXTocNEET_v2020; Rödenbeck et al., 2005). The GCP inversion
protocol recommends to use as a fixed prior the same gridded dataset of $FCO_2$ emissions (GCP-GridFED).
However, some modelers used different interpolations of this dataset, and one group used a different gridded





dataset (Ciais et al., 2021). We applied a correction to the estimated total $CO_2$ flux by subtracting a common
$FCO_2$ flux from each inversion (Figure S8 and Methodological Supplementary 2). The resulting land
atmosphere $CO_2$ fluxes, or net ecosystem exchange, cannot be directly compared with inventories aiming
to assess C stock changes, given the existence of land-atmosphere $CO_2$ fluxes caused by lateral processes.
This issue was discussed by Ciais et al. (2021) and a practical correction of inversions was proposed by
Deng et al. (2022) based on new datasets for $CO_2$ fluxes induced by lateral processes involving river
transport, crop and wood product trade. We applied here the same correction to all $CO_2$ inversions.
**$CH_4$ inversions**
We used the $CH_4$ emissions from global inversions over 2000-2017 from the Global Methane Budget
(Saunois et al., 2020) (Table 1). This ensemble includes 11 models using GOSAT satellite $CH_4$ total-column
observations covering 2010-2017, and 11 models assimilating surface stations data (SURF) since 2000
(Table S4). Surface inversions are constrained by very few stations for Africa, while the GOSAT satellite
data has a better coverage. One could thus expect GOSAT inversions to give more robust results. Inversions
deliver an estimate of surface net $CH_4$ emissions, although some of them solve for fluxes in groups of
sectors, called 'super-sectors'. In the inversion dataset, net $CH_4$ surface emissions were interpolated into a
$0.8° \times 0.8°$ resolution, regridded from coarser resolution fluxes and separated into 'super-sectors' either
using prior emission maps or posterior estimates for those inversions solving fluxes per supersector,
following Saunois et al., (2020). More specifically, these five super-sectors are: 1) Fossil Fuel, 2)
Agriculture and Waste, 3) Wetlands, 4) Biomass and Biofuel Burning (BBUR), and 5) Other natural
emissions. We separated $CH_4$ anthropogenic emissions from inversions using Method 1 and Method 2
proposed by Deng et al. (2021). Method 1 relies on the separation calculated by each inversion except for
the BBUR supersector from which wildfire emissions were subtracted based on the Global Fires Emission
Dataset (GFED) version 4 (van der Werf et al., 2017). Method 2 removes from total emissions the median
of natural emissions from inversions (Deng et al. 2022). The two methods gave similar results and only
Method 1 was used in the results section.
**$N_2O$ inversions**
We used three $N_2O$ atmospheric inversions from the global $N_2O$ budget synthesis (Tian, 2020) and from
Deng et al. (2022) ( Tables S1, S7) : PyVAR CAMS (Thomson et al., 2014), MATCM_JMASTEC
(Rodgers, 2000), (Patra et al., 2018), and TOMCAT (Wilson et al., 2014; Monks et al., 2017). We use the
total $N_2O$ flux from inversions including natural emissions, given that natural emissions estimates are highly
uncertain for Africa. Inversion results are therefore not directly comparable with the PRIMAP-hist inventory
which only contains anthropogenic emissions.



**1.5 Metrics to compare gasses and ancillary data**

We express emissions of non-$CO_2$ gasses in megatons of carbon dioxide equivalent (MtCO$_2$e) using the Global Warming Potential over a 100-year time horizon (GWP100) values from the fourth IPCC Assessment Reports (IPCC AR4, WGI Chapter 2, 2007), consistent with PRIMAP-hist and country-reported data. We used population data from the United Nations population (World Population Prospects 2019: Highlights | Multimedia Library - United Nations Department of Economic and Social Affairs, 2022), for computing per capita $FCO_2$ emissions and their disparities, based on Gini indices (Dortman et al.,1979) for measuring statistical dispersions among a given population (methodological supplementary M1). We also used African GDP data (World Bank, 2017).

**2 Results and discussion**

**2.1 Fossil $CO_2$ emissions**

**2.1.1 Continental, regional and country changes**

**(a)**

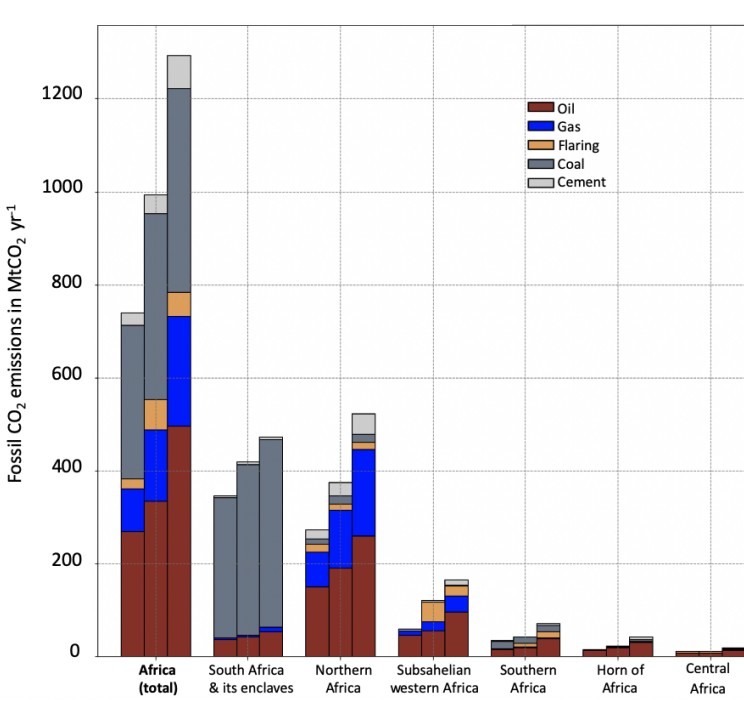

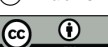

**(b)**

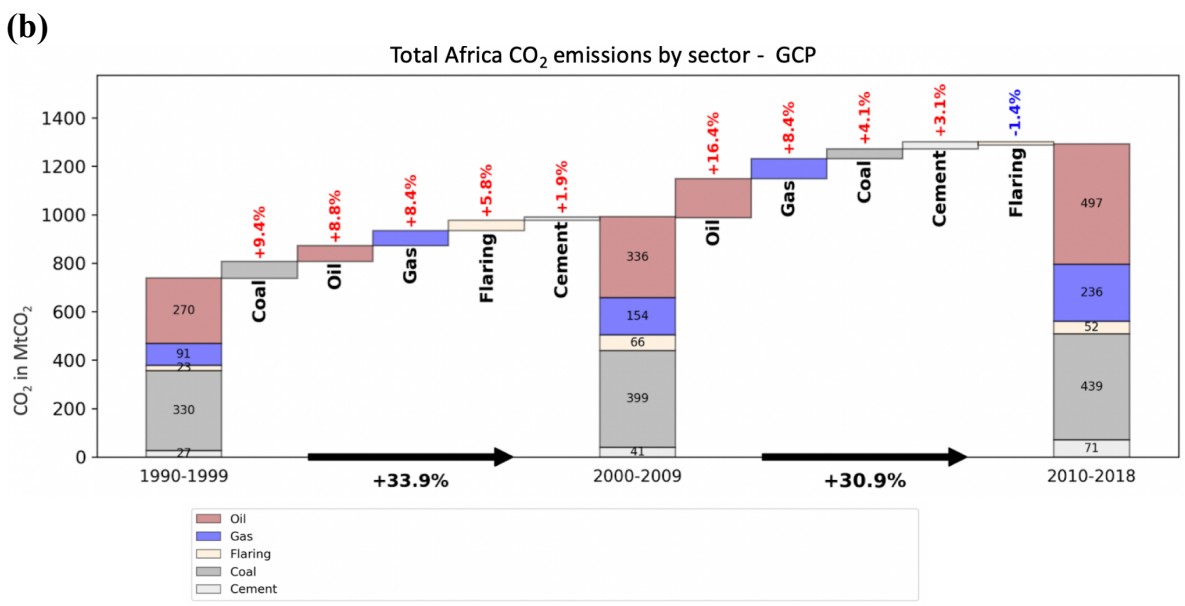

**(c)**

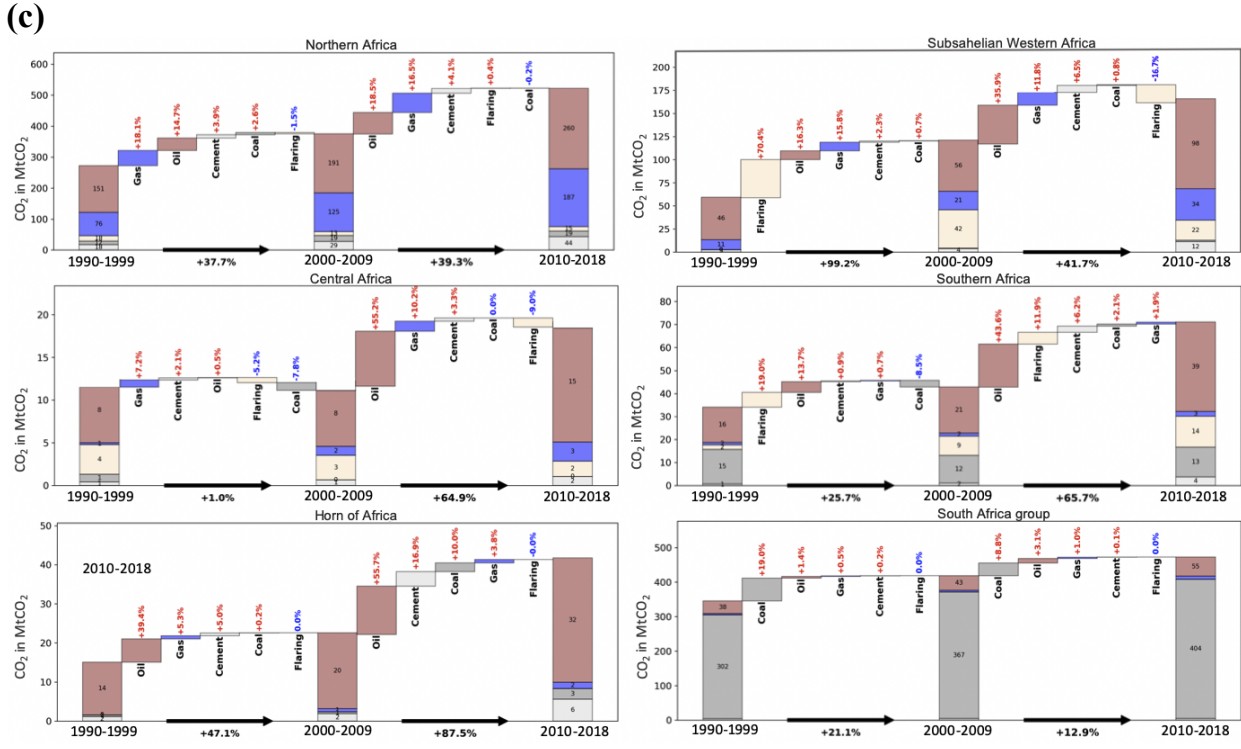



**Figure 2. (a) African fossil fuel CO₂ emissions per fuel type and for cement per region over 1990-1999, 2000-2009 and 2010-2018. (b) Contribution of each fuel type to the change of African emissions. (c) Same for different regions regrouping several countries. Data from GCP (2019).**

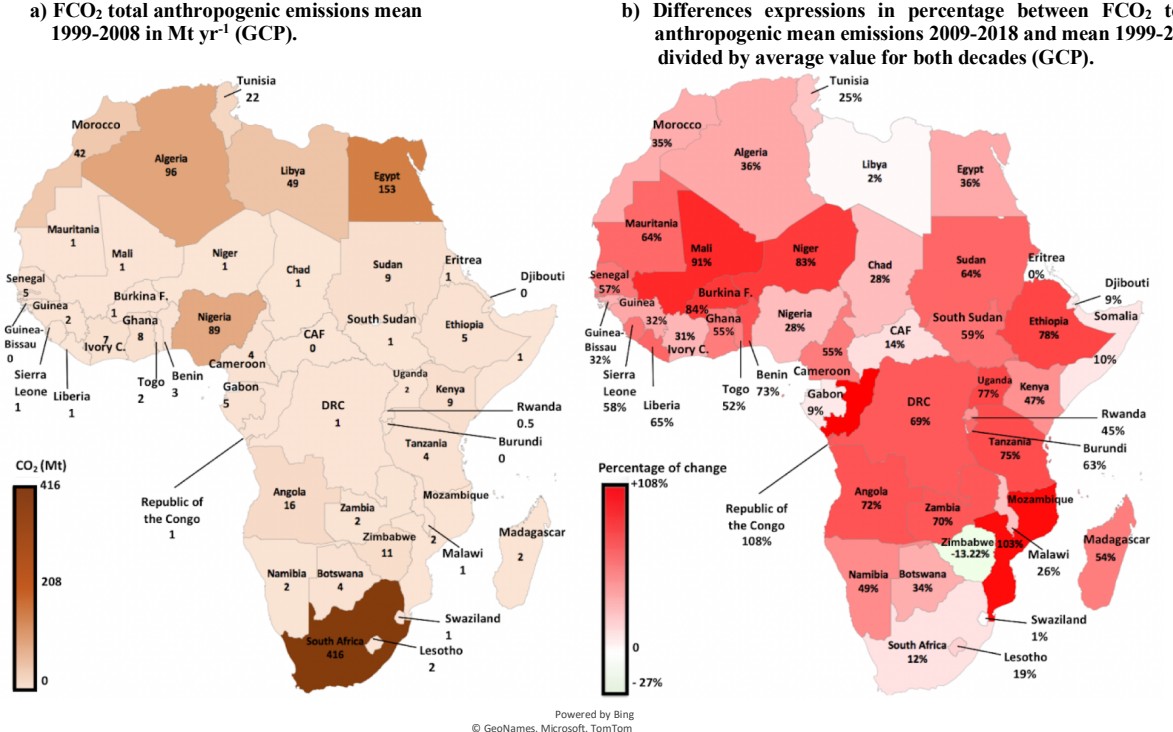



**c) CH₄ total anthropogenic emissions mean 1999-2008 in MtCO₂e yr⁻¹ (PRIMAP-hist).**

**d) Differences between CH₄ total anthropogenic mean emissions 2009-2018 and mean 1999-2008 in MtCO₂e yr⁻¹ (PRIMAP-hist).**

**e) N₂O total anthropogenic emissions mean 1999-2008 in MtCO₂e yr⁻¹ (PRIMAP-hist).**

**f) Differences between N₂O total anthropogenic mean emissions 2009-2018 and mean 1999-2008 in MtCO₂e yr⁻¹ (PRIMAP-hist).**



**Figure 3 (a). Maps of average fossil fuel CO₂ emissions for African countries during 1999-2008 in MtCO₂e yr⁻¹ and**
**(b) change from 1999-2008 to 2009-2018 using data from GCP in MtCO₂e yr⁻¹ (Friedlingstein et al., 2019); (c-d)**
**same but with anthropogenic CH₄ emissions from PRIMAP-hist in MtCO₂e yr⁻¹; (e-f) same for anthropogenic N₂O**
**emissions from PRIMAP-hist in MtCO₂e yr⁻¹.**
**PRIMAP-hist and GCP**
First, we compared GCP and PRIMAP-hist fossil $CO_2$ emissions. We found that most of the relative differences
between these two datasets at country level considerably decreased with time, except for Mali. Those
differences are less than 5% for most of the main African emitters during the last decade, except for South
Africa where the difference is a bit larger than 10% (see maps in Fig. S7). The largest relative difference
between the two datasets comes from Mali in the decade 2009-2018, with $FCO_2$ emissions of 3 $MtCO_2$ $yr^{-1}$ in
GCP, compared to 1 $MtCO_2$ $yr^{-1}$ in PRIMAP-hist. Given the relatively small differences, we chose to use only
GCP for trends between decades, but when computing net budgets for the three main GHG, we show
differences between the use of those two estimates.
The changes of African $FCO_2$ emissions per fuel type and for cement using the GCP data are shown in Fig. 2
(a). In Fig.2 (b), we show absolute values and relative contributions to the total change in each decade. During
2010-2018, total African $FCO_2$ emissions from oil (497 $MtCO_2$ $yr^{-1}$) and coal (439 $MtCO_2$ $yr^{-1}$) were roughly
similar. While global $FCO_2$ emissions increased by +13 % over this period (Friedlingstein et al., 2019), African
$FCO_2$ almost doubled in 2018 compared to 1990 levels, a relative increase comparable with that of China over
the same period. From 1990-1999 to 2000-2009, the mean emissions increased by 33.9% from 741 $MtCO_2$ $yr^{-}$
$^1$ to 996 $MtCO_2$ $yr^{-1}$. All $FCO_2$ sectors contributed to this decadal increase. The contribution from coal (+9.4
%) was slightly larger but comparable to that from oil (+9 %) and gas (+8 %). From 2000-2009 to 2010-2018,
emissions further increased by 31% from 996 $MtCO_2$ $yr^{-1}$ to 1295 $MtCO_2$ $yr^{-1}$. The oil and the gas fuels
contributed the most to this increase with +16 % for oil, and +8 % for gas. Coal emissions increased by only
+4.1 % and coal went from being the first source of African $FCO_2$ emissions over 2000-2009 to the second one
over 2010-2018.
As for regional contributions to emissions changes between 1990-1999 and 2000-2009 shown in Fig. 2 (b) the
main contribution to the total increase came from the region of South Africa where emissions increased from
302 $MtCO_2$ $yr^{-1}$ to 367 $MtCO_2$ $yr^{-1}$ (+21.1 %, coal being the largest contributor). The second largest contribution
to the increase is from North Africa where oil was the largest contributor (emissions increased from 151 $MtCO_2$
$yr^{-1}$ to 191 $MtCO_2$ $yr^{-1}$; +15 %), and gas (+18%). The least increasing region was Central Africa. North Africa
experienced the largest increase from 1990-1999 to 2000-2009, and from 2000-2009 to 2010-2018 with





successive increases of +38 % and +39 %, largely dominated by oil and gas (Fig. 4 (b)). As a result, during the
period 2010-2018, Northern African countries were the dominant emitters with 545 $MtCO_2$ $yr^{-1}$. The group of
South Africa (including Lesotho and Botswana) was the second biggest emitter region over 2010-2018, mainly
due to coal emissions from the Republic of South Africa. The two least contributing African regions were the
Horn of Africa and Central Africa.
At the country level, Figure 3a-b shows mean $FCO_2$ emissions and relative changes over the last two decades.
The main emitters do not have the biggest relative changes. The four main emitters over 2000-2009 were South
Africa (416 $MtCO_2$ $yr^{-1}$), Egypt (153 $MtCO_2$ $yr^{-1}$), Algeria (96 $MtCO_2$ $yr^{-1}$) and Nigeria (89 $MtCO_2$ $yr^{-1}$). Those
four countries altogether represented 67% of the continental total emissions over 2000-2009 (987 $MtCO_2$ $yr^{-1}$).
The largest relative increases from 2000-2009 to 2010-2018 are from Congo (+108 %), Mozambique (+103 %)
and Mali (91%), compared to relative increases in the main emitters, the Republic of South Africa (+21 %),
Egypt (+36%) and Algeria (+36%).
**2.1.2 Variations of per capita and per GDP fossil fuel $CO_2$ emissions**
**Per capita emissions**
Using ancillary data on population (Fig. S2 and Fig. S3) we computed the mean African per capita emissions
of 1 $tCO_2$/cap $yr^{-1}$ for 2009-2018 (Figures S2 and S3), which is 5 times larger than during 1990-1998 (0.2 tC/cap
$yr^{-1}$), and yet 5 times smaller than the global average (5 $tCO_2$/cap $yr^{-1}$). From 1999-2008 to 2009-2018, African
per capita emissions increased by 30 %. African per capita $FCO_2$ emissions during 2009-2018 were 17 times
less than in the USA (17 $tCO_2$/cap $yr^{-1}$), 7 times less than in China (7 $tCO_2$/cap $yr^{-1}$), 7 times less than in
EU27+UK (7 $tCO_2$/cap $yr^{-1}$), and 2 time less than India (2 $tCO_2$/cap $yr^{-1}$). At the country level, the biggest per
capita emissions over 2009-2018 were from the Republic of South Africa with 9 $tCO_2$/cap $yr^{-1}$, which ranks
14[th] worldwide, above China and just below Poland. The second biggest per capita emissions were from Libya
(8 $tCO_2$/cap $yr^{-1}$). The smallest ones were from the DRC (0.1 $tCO_2$/cap $yr^{-1}$). For the first period 1990-1998,
per capita emissions of African region ranked in this order: South Africa group (4 $tCO_2$/cap $yr^{-1}$) > Northern
Africa (2 $tCO_2$/cap $yr^{-1}$) > Central African countries (1 $tCO_2$/cap $yr^{-1}$) > Southern countries (0.8 $tCO_2$/cap $yr^{-1}$)
> Horn of Africa (0.5 $tCO_2$/cap $yr^{-1}$) > Sub-Sahelian Western Africa (0.3 $tCO_2$/cap $yr^{-1}$). For the second period
2009-2018, they ranked in this order: South Africa group (4 $tCO_2$/cap $yr^{-1}$) > Northern Africa (2 $tCO_2$/cap $yr^{-1}$)
> Southern countries (1 $tCO_2$/cap $yr^{-1}$) and Horn of Africa (1 $tCO_2$/cap $yr^{-1}$) > Central Africa countries (1
$tCO_2$/cap $yr^{-1}$) > Sub-Sahelian Western Africa (0.4 $tCO_2$/cap $yr^{-1}$). At country scale during the first period of
1990-1998, the four African largest per capita emissions ranked in this order: Libya (9 $tCO_2$/cap $yr^{-1}$ > the
Republic of South Africa (9 $tCO_2$/cap $yr^{-1}$) > Gabon (5 $tCO_2$/cap $yr^{-1}$) > Algeria (3 $tCO_2$/cap $yr^{-1}$). The four



African countries with the smallest per capita emissions ranked as following: Burundi (0.04 $tCO_2$/cap $yr^{-1}$) <
Uganda, Ethiopia and Mali (0.1 $tCO_2$/cap $yr^{-1}$).
We also computed the GINI index for African per capita $FCO_2$ emissions for each of the last three decades,
using data from (Friedlingstein et al., 2020) (see Methodological Supplementary M2). These GINI values were
0.7 for 1990-1998, 0.7 for 1999-2008, and 0.7 for 2009-2018, thus very stable over the last 30 years and close
to 1, indicating high inequities among countries.
**Emissions per GDP**
**Per exchange rate vs. per Purchasing Power Parities (PPP) GDP**
According to the International Monetary Fund (IMF), the Gross Domestic Product (GDP) delivers an estimate
"of the monetary value of goods and services produced in a country over a chosen period." GDP data from the
World Bank (2015) is available for 30 African countries only (Fig. S4). The four countries with the biggest per
$US exchange rate GDP (Fig. S5) are: Nigeria ($490 B) > South Africa ($350B) > Egypt ($330B) and Algeria
($330B) > Angola ($120B). The four countries with the smallest GDP in 2015 are: Gambia ($1.4B) and
Seychelles ($1.4B) > Guinea-Bissau ($1B) > Comoros ($970 M). Emissions per $US GDP are shown in Fig.
S5 The Purchasing Power Parities (PPP) calculated by the International Comparison Program (ICP) of the
World Bank is a refined measure of what a given national currency can acquire in terms of goods or services
in another country, removing the impact of currency exchange rates. Emissions per PPP$ GDP are shown in
Fig. S6.
The mean of African emissions per unit PPP$ GDP in 2016 was 0.6 $kgCO_2$/PPP$ $yr^{-1}$, which is more than twice
the global value, 3 times the mean value of the USA (0.2 $kgCO_2$/PPP$ $yr^{-1}$) and Europe (0.2 $kgCO_2$/PPP$ $yr^{-1}$
$^{1}$). This points to a more carbon intensive economic growth in Africa than in developed countries, which may
be an important barrier for future mitigation strategies as the GDP of Africa has grown by 112% in the last 30
years, and is projected to increase in the future by 3% per year (World Bank, 2022). At regional level, the
largest values were: South Africa (0.4 $kgCO_2$/PPP$ $yr^{-1}$) > North Africa, Southern Countries and Sahelian
Western Africa (0.2 $kgCO_2$/PPP$ $yr^{-1}$) > Central Africa and the Horn of Africa (0.1 $kgCO_2$/PPP$ of GDP). At
country scale, the largest emitters per unit of GDP were Libya (0.7 $kgCO_2$/PPP$ $yr^{-1}$) and South Africa (0.7
$kgCO_2$/PPP$ $yr^{-1}$) > Lesotho (0.4 $kgCO_2$/PPP$ $yr^{-1}$) > Algeria (0.3 $kgCO_2$/PPP$ $yr^{-1}$) (Fig. S6.) The smallest
emitters were: DRC (0.03 $kgCO_2$/PPP$ $yr^{-1}$) < Chad (0.04 $kgCO_2$/PPP$ $yr^{-1}$) < Burundi (0.06 $kgCO_2$/PPP$ $yr^{-1}$
$^{1}$) < Uganda (0.07 $kgCO_2$/PPP$ $yr^{-1}$).
We also used GDP per unit exchange rate from the International Energy Agency (IEA, 2019). The mean African
emissions per unit of GDP$_{exch.rate}$ was 0.5 $kgCO_2$ $/$ $yr^{-1}$, larger than elsewhere, except in Asia (0.6
$kgCO_2$/GDP$_{exch.rate}$ $yr^{-1}$. As shown in Fig. S5, over 2013-2017 the six biggest emitters were South Africa (0.7



kgCO$_2$/GDP$_{exch.rate}$ yr$^{-1}$) > Libya (0.5 kgCO$_2$/GDP$_{exch.rate}$ yr$^{-1}$) > South Sudan (0.4 kgCO$_2$/GDP$_{exch.rate}$ yr$^{-1}$)
> Zimbabwe, Benin and Algeria (0.3 kgCO$_2$/GDP$_{exch.rate}$ yr$^{-1}$). The correlation coefficient between
GDP$_{exch.rate}$ and FCO$_2$ emissions per GDP$_{exch.rate}$ was 0.3, suggesting that the countries with a high GDP do
not always emit more CO$_2$ per unit GDP. For instance, South Africa ranked first with 0.7 kgCO$_2$/GDP$_{exch.rate}$
yr$^{-1}$ and second for GDP (350 $Billion); Nigeria ranked first for GDP (490 $Billion), but 21$^{st}$ for emissions per
GDP (0.1 kgCO$_2$/GDP$_{exch.rate}$ yr$^{-1}$). This may be related to the fact that countries with a high GDP are also
more likely to create growth through sustainable activities.
**2.2 LULUCF CO$_2$ fluxes**

453        **Outlier corrections**

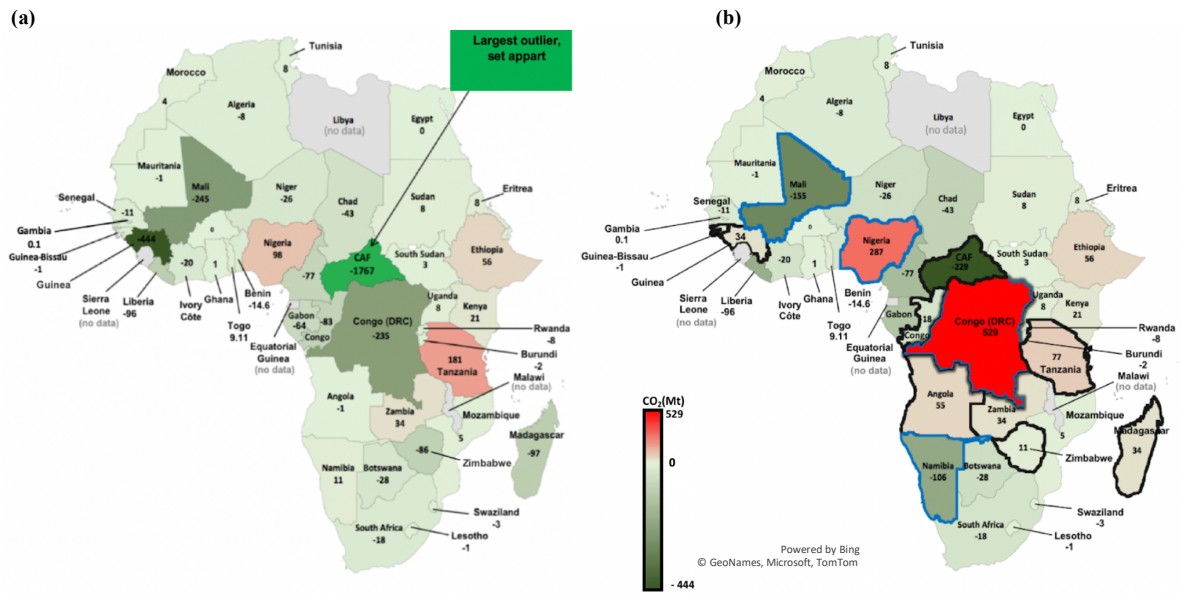

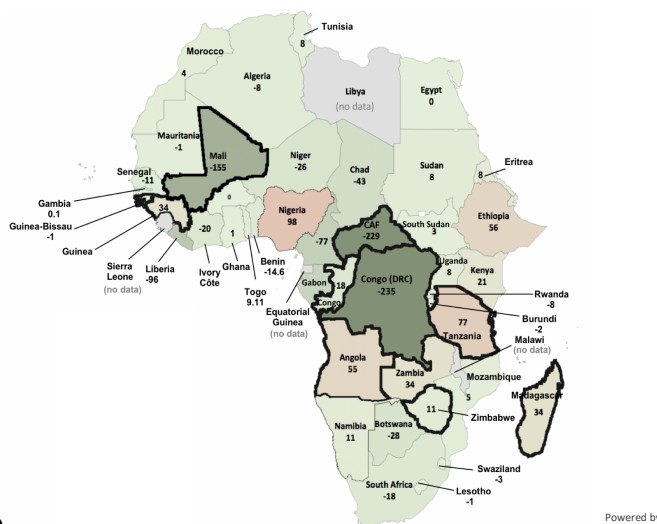

**(c)**

**Figure 4. Map of national LULUCF CO$_2$ fluxes for 2001-2018 in MtCO$_2$ yr$^{-1}$. (a) before outliers' removals. ((b) After outliers' removal according to Grassi et al. (2022). (c) After outlier removals (DRC, Namibia and Nigeria) from this study. Positive values represent a net C loss by ecosystems.**

In this section, we analyze CO$_2$ fluxes from the LULUCF sector, based on UNFCCC data (section 1.1) which include forest lands, grasslands, croplands, and all possible conversions between them (IPCC, 2003; 2006). As shown in section 1.2 and Table S4, we found that some countries' reports are outliers with biophysically implausible CO$_2$ sinks and/or sudden unexplained very large changes between successive reports. Due to scarce data over 1990-1998 we focus on the period 2001-2018. In the following paragraph, we discuss four approaches to include UNFCCC data:

a) Uncorrected data, b) corrections following Grassi et al. (2022) for all countries, c) corrections following Grassi et al. (2022) except DRC, Namibia and Nigeria, d) Corrections following Grassi et al. (2022) except DRC.

Figure 4 (a) shows UNFCCC data without correcting for outliers, based on BUR and NC data accessed in May 2022. The majority of countries are sinks, or small sources, except Tanzania and Nigeria being large sources. Very large (implausible) sinks are seen in Guinea and CAF. The continent is a CO$_2$ sink of -3309 MtCO$_2$ yr$^{-1}$ during the period 2001-2018.

Figure 4 (b) shows the corrected fluxes according to Grassi et al. (2022) who excluded implausible large sink rates and used NDC and REDD+ reports instead of NC data for DRC, Congo, CAF, Guinea, Madagascar and the most recent BUR, NC and inventory data for Namibia, Angola, Zimbabwe and Nigeria (see their Table 7). Africa as a whole is a CO$_2$ source of 265 MtCO$_2$ yr$^{-1}$. At regional scale, the mean CO$_2$ sources distributes as follows on four regions: Sub-Sahelian West Africa (235 MtCO$_2$ yr$^{-1}$) > Horn of Africa (153 MtCO$_2$ yr$^{-1}$) > Central Africa (144 MtCO$_2$ yr$^{-1}$) > Southern Africa (14 MtCO$_2$ yr$^{-1}$).



The two sink regions are North Africa (-259 $MtCO_2$ $yr^{-1}$) and South Africa (-23 $MtCO_2$ $yr^{-1}$). At country
scale, after the corrections of Grassi et al. (2022), the four countries with the larger sinks are: CAF (-229
$MtCO_2$ $yr^{-1}$) > Mali (-155 $MtCO_2$ $yr^{-1}$) > Namibia (-106 $MtCO_2$ $yr^{-1}$) > Cameroon (-77 $MtCO_2$ $yr^{-1}$). The
four countries with largest sources are DRC (529 $MtCO_2$ $yr^{-1}$) > Nigeria (287 $MtCO_2$ $yr^{-1}$) > Tanzania
(77 $MtCO_2$ $yr^{-1}$) > Ethiopia (56 $MtCO_2$ $yr^{-1}$). A main issue with the correction from Grassi is that it reports
no sink in DRC which has an important forest coverage representing 68% of the country area (FAO,
2015) and for which a sink was consistently reported in previous NCs.
Figure 4 (c) shows LULUCF $CO_2$ in African countries that are consistent with Grassi et al. (2022) except
for three countries: Namibia (we used 2000 NC3 instead of NIR2019), Nigeria (we used 2014 NC2
instead of 2017 BUR2) and DRC (we used 2015 NC3 instead of 2021 NDC). In that approach Africa
becomes a net $CO_2$ sink of -589 Mt $yr^{-1}$ over 2001-2018. At regional scale, the region of Central Africa
(-620 $MtCO_2$) remains the main sink. But the values and ranking of the top sources rank as: Horn of
Africa (153 $MtCO_2$) > Southern Africa (141 $MtCO_2$) > Sub-Sahelian West Africa (19$MtCO_2$). At country
scale with this correction choice, the top sinks are: DRC (-235 $MtCO_2$) > CAF (-229 $MtCO_2$) > Mali (-
155 $MtCO_2$); and the three top sources: Nigeria (98 $MtCO_2$) > Tanzania (77 $MtCO_2$) > Ethiopia (56
$MtCO_2$).
In the fourth approach where we use the corrections of Grassi et al. (2022) except for DRC where we
kept the latest national communication instead of the most recent NDC, the continent is a net sink of -
504 $MtCO_2$ $yr^{-1}$ over 2001-2018. At regional scale, Central Africa is a large $CO_2$ sink, and the ranking
of sink regions is: Central African group (-620 $MtCO_2$ $yr^{-1}$) > North Africa (-259 $MtCO_2$ $yr^{-1}$) > South
Africa (-23 $MtCO_2$ $yr^{-1}$). The ranking of the source regions stays unchanged. At the country scale, the
main sink is DRC (-235 $MtCO_2$ $yr^{-1}$). In the paper, we will mainly use data corrected following Grassi et
al. (2022), but we want to raise a caution flag that adopting their correction for DRC had an enormous
effect on the $CO_2$ budget of the continent, which becomes a source. Using the original latest national
communication of DRC instead of the NDC used by Grassi et al., and our own corrections for Namibia
and Nigeria instead of those of Grassi et al. increased the continental $CO_2$ uptake.
**Comparison of UNFCCC managed land area and FAO forest and other wooded lands areas**
Figure S10 shows a comparison of land areas reported in NC, BUR and REDD+ reports
(https://redd.unfccc.int/submissions.html?mode=browse-bycountry) with FAO forest land areas (2015)
and FAO forest land + other woodlands areas for the year 2015 (see Table S8). Consistent with Grassi
et al. (2022), all forest lands in Africa are considered as managed. We found that FAO forest lands areas
are closer to UNFCCC estimates than the sum of FAO forest and other woodlands area, except for DRC,





Sudan, Senegal, Niger and Mauritania (Table S8). Forest areas in UNFCCC data using IPCC default
method do not exactly match FAO data estimates of forest area.
**LULUCF CO$_2$ fluxes from UNFCCC versus DGVM and inversions**
A comparison between LULUCF CO$_2$ fluxes from UNFCCC, FAO, DGVMs and inversions is shown
in Fig. 5 at the scale of the continent and for the six regions. The period of overlapping time series is
2001-2018. For the continent, DGVMs give a mean sink of -232 MtCO$_2$ yr$^{-1}$ with a huge range from -
1977 MtCO$_2$ yr$^{-1}$ to 2095 MtCO$_2$ yr$^{-1}$. The years with the biggest sinks for DGVM (from the median of
all models) are 2006 and 2018, and the years with the smallest sinks are 2005 and 2016 which seem
related to widespread drought years across Africa. A key result shown by this figure is that the DGVMs
and inversions show a huge spread, making them of little value to 'verify' inventories for LULUCF
CO$_2$ fluxes in Africa. Yet, we observed that the median of all DGVM points to a sink for Africa, unlike
the UNFCCC data with the correction from Grassi et al. (2022).

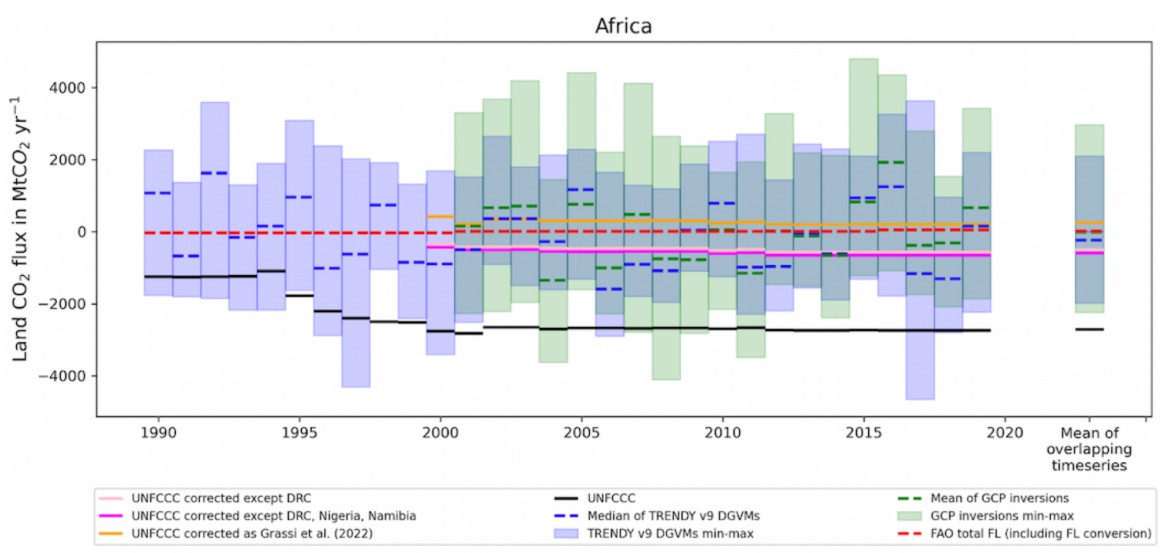


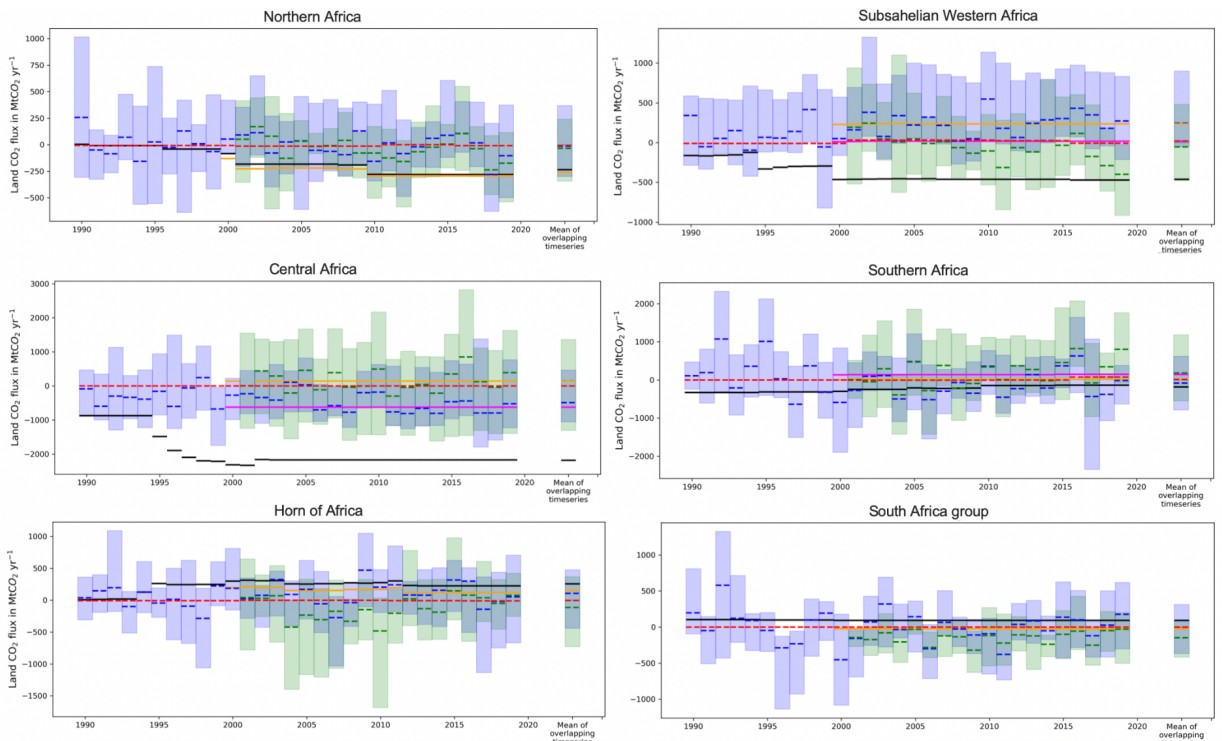

**Figure 5. LULUCF CO₂ emissions and sinks: comparison between UNFCCC national greenhouse gas inventories, TRENDYv9 DGVMs and inversions, for total Africa and for each of the six African sub regions; as well as country details for the three lain outliers. The unit is in MtCO₂ yr⁻¹ Shaded green areas represent the minimum and maximum ranges from inversions. Shaded blue represents the minimum and maximum ranges for TRENDYv9 DGVMs. Green dashes denote the mean of inversions, blue dashes denote the median of TRENDYv9 DGVMs, green dashes the median of inversions. Positive values represent a source while the negative values refer to a sink.**

For three large countries, corrected UNFCCC values from Grassi et al. show a bigger discrepancy with other BU and TD methods than uncorrected ones (Fig. S9). In Namibia the corrected value gives a larger sink compared to other methods, while the uncorrected value is comparable. In DRC the corrected value which was a source seems a high overestimate compared to other methods, while the uncorrected UNFCCC value is close to median values from inversions, and to FAO. In Nigeria, the corrected value seems to be a high overestimation of a net source compared to other methods pointing to either a smaller source (FAO, inversions) or even a sink (DGVM).

The data in Figure 5 show that most methods agree on a small net sink for African LULUCF CO₂ fluxes, except for corrections following Grassi et al (2022). But disagreements exist among different methods.


Inversions give a smaller net sink ($\text{mean}_{\min}^{\max}$) of -14 $_{-2\,248}^{2\,966}$ MtCO$_2$ yr$^{-1}$ than DGVMs (-232 $_{-1978}^{2095}$ MtCO$_2$
yr$^{-1}$). The median value of inversions is nevertheless within the range of DGVMs. At the scale of Africa,
the inversions mean sink is ~12 times smaller than the median from DGVMs. The min-max range of
inversions (5216 MtCO$_2$ yr$^{-1}$) is larger than the range of the DGVMs by 17%. DGVM and inversions show
a positive temporal correlation coefficient (r = 0.7) for annual trends (linear fit to time series).
UNFCCC values with the fourth approach point out to a net sink (-503 MtCO$_2$ yr$^{-1}$), similar to the third
one. Corrected values as in Grassi et al. (2022) give a net source estimate of 265 MtCO$_2$ yr$^{-1}$. FAO net
emissions and removals represent a small net source (18 MtCO$_2$ yr$^{-1}$). Differences between FAO and
UNFCCC, as explained in Grassi et al. (2022), could be due to the fact that FAO estimates of CO$_2$ fluxes
for forest remaining forest can be set to zero in absence of any national stock change inventory (Table 3).
**Table 3. Mean net LULUCF CO$_2$ (emissions and removals) over the overlapping period of the different**
**datasets (2001-2018), in MtCO$_2$ yr$^{-1}$.**

| Region | Corrected UNFCCC (Grassi et al. 2022) with and without DRC correction. | Corrected UNFCCC but DRC/ Nigeria/ Namibia | Median TRENDY v9 | Max TRENDY v9 | Min TRENDY v9 | Mean GCB inv. | Max GCB inv. | Min GCB inv. | FAO total FL with FL conversion |
|---|---|---|---|---|---|---|---|---|---|
| South Africa group | -23 | -23 | -5 | 312 | -368 | -147 | 96 | -418 | -1 |
| Horn of Africa | 153 | 153 | 108 | 475 | -439 | -115 | 367 | -729 | -5 |
| Southern Africa | 14 | **141** | -81 | 622 | -785 | 182 | 1186 | -548 | 13 |
| North Africa | -259 | -259 | -13 | 369 | -299 | -34 | 240 | -343 | -9 |
| Subsahelian West Africa | 236 | **19** | 245 | 900 | -49 | -53 | 481 | -479 | 21 |
| Central Africa | **144** (DRC with NDC2021) **-620** (DRC with NC3) | -620 | -490 | 461 | -1051 | 152 | 1362 | -1303 | -1 |
| Africa total | **265** (DRC with NDC2021) **-503** (DRC with NC3) | -589 | -232 | 2095 | -1978 | -14 | 2967 | -2249 | -1 |

At a regional scale, we note some agreement between different bottom-up approaches. First, for the South
Africa region, the mean of DGVM medians during the overlapping period 2001-2018 (-5 MtCO$_2$ yr$^{-1}$) and
the FAO estimate (-1 MtCO$_2$ yr$^{-1}$) are comparable and not too far from Grassi et al., 2022 (-23 MtCO$_2$ yr$^{-}$
$^1$). Second, for North Africa, the DGVM median (-13 MtCO$_2$ yr$^{-1}$) and the FAO mean estimate over the



same period (-9 MtCO$_2$ yr$^{-1}$) are comparable. Finally, in Sub-Sahelian West Africa, the DGVM (236
MtCO$_2$ yr$^{-1}$) and UNFCCC corrected following Grassi et al., 2022 (245 MtCO$_2$ yr$^{-1}$) are also close to each
other.
Northern Africa is the group where DGVM and inversions point to the closest values both in terms of sign
(sink) and magnitudes with respectively small sinks of $-13^{369}_{-299}$ MtCO$_2$ yr$^{-1}$ and $-34^{240}_{-343}$ MtCO$_2$ yr$^{-1}$.
Looking at DGVM and inversions in the region of South Africa, we found that both DGVM and inversions
point to a sink (respectively $-5^{312}_{-368}$ MtCO$_2$ yr$^{-1}$ and $-147^{96}_{-418}$ MtCO$_2$ yr$^{-1}$), however with a different
magnitude. The region showing the highest discrepancies between inversions and DGVM values is Central
Africa with a source in inversions ($152^{1362}_{-1303}$ MtCO$_2$ yr$^{-1}$) and a sink for DGVM ($-490^{461}_{-1051}$ MtCO$_2$ yr$^{-1}$).
The Sub-Sahelian West Africa also shows discrepancies in both sign and magnitude with $245^{900}_{-49}$ MtCO$_2$
yr$^{-1}$ for DGVM and $-53^{481}_{-479}$ MtCO$_2$ yr$^{-1}$ for inversions. The same is true for Southern Africa with
$-81^{622}_{-785}$ MtCO$_2$ yr$^{-1}$ for DGMVs and $182^{1186}_{-548}$ MtCO$_2$ yr$^{-1}$ for inversions, and the Horn of Africa
with $108^{475}_{-439}$ MtCO$_2$ yr$^{-1}$ for DGVM and $-115^{367}_{-729}$ MtCO$_2$ yr$^{-1}$ for inversions. At the regional scale, the
inversions systematically give smaller sinks than DGVMs in the regions of Central Africa, Sub-Sahelian
West Africa and North Africa after 2010 (Fig. 5).
We also computed the coefficient of correlation at the regional level between DGVM and inversions trends
for each region. The highest correlation coefficients are in the South Africa region (r = 0.7), followed by
Northern Africa (r = 0.6) and in Southern Africa (r = 0.5). The lowest correlation coefficients are for the
group of Central African countries (r = 0.3), Sub-Sahelian Western countries (r = 0.2) and the Horn of
Africa (r = 0.1).
**2.3 CH$_4$ anthropogenic emissions**
**Total and sectoral bottom up CH$_4$ anthropogenic emissions and decadal changes**

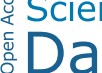

**(a)**

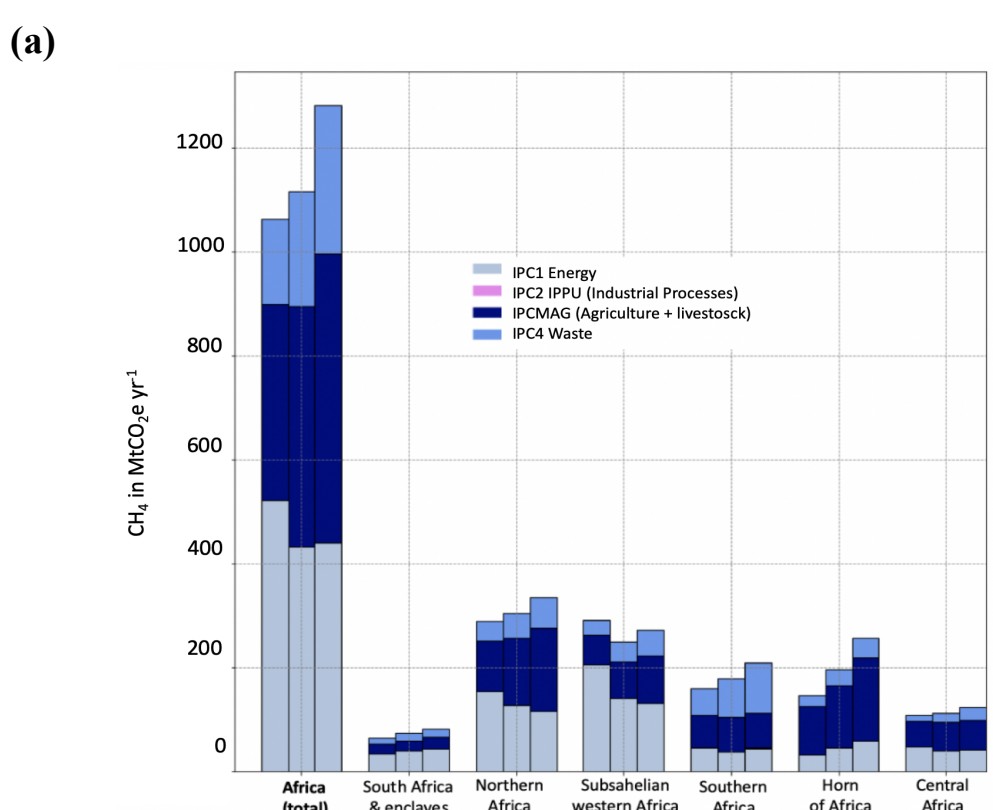

**(b)**

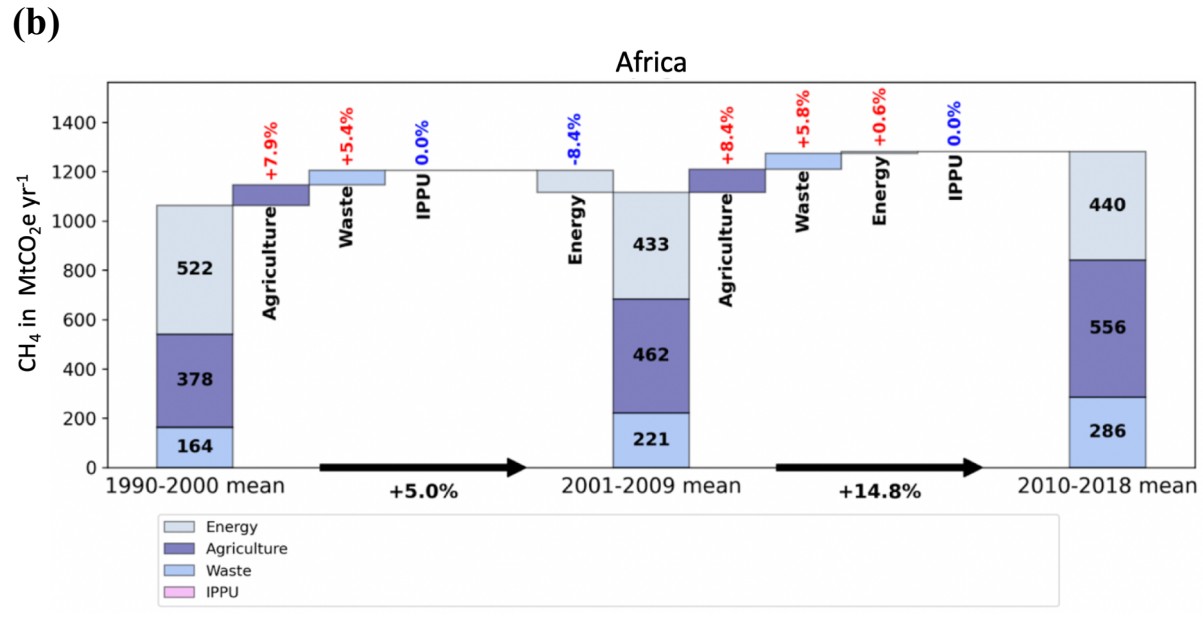

**(c)**

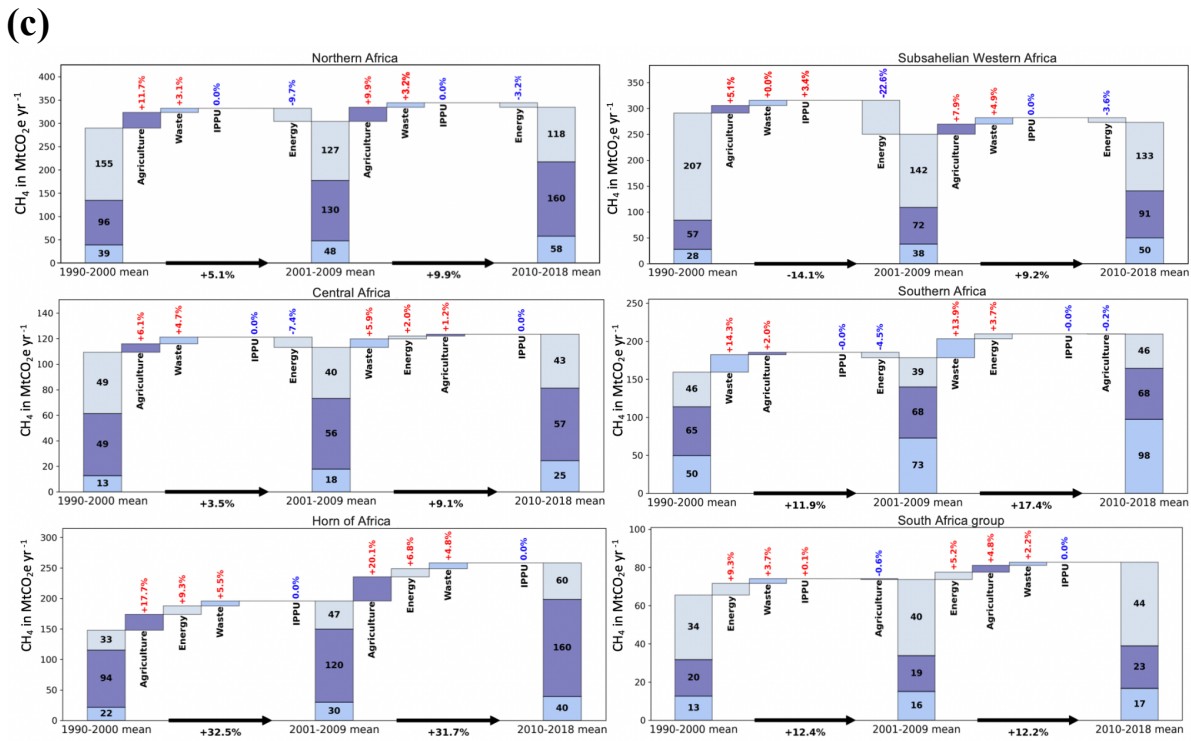

**Figure 6. (a) African mean anthropogenic CH₄ emissions in MtCO₂e yr⁻¹ over three decades (1990-1998, 1999-2008,**
**2010-2018). (b) Contribution of each sector to the change of African emissions between the last three decades. (c)**
**Same for different regions regrouping several countries. Data from PRIMAP-hist (2021).**
Figure 6 shows anthropogenic CH₄ emissions from PRIMAP-hist grouped into four super-sectors (see section
1). A map of CH₄ emissions and their trends per country is given in Fig. 3c-d. LULUCF CH₄ emissions are not
considered in PRIMAP-hist. African anthropogenic CH₄ emissions sum up to 1154 MtCO₂e yr⁻¹ over the last
three decades. They increased from 1064 MtCO₂e yr⁻¹ in 1990-2000 to 1116 MtCO₂e yr⁻¹ in 2001-2009, and
further to 1282 MtCO₂e yr⁻¹ over 2010-2018 (Fig. 6.a.) Over the last three decades, the main African CH₄
emitting super-sectors shifted from Energy (49% over 1990-2000) to Agriculture, mainly due to a North
African contribution. At the regional level, the main contributing region to total emissions shifted over the last
30 years from Sub-Sahelian Western Africa (297 MtCO₂e yr⁻¹ for all sectors in 1990-2000) to North Africa
(333 MtCO₂e yr⁻¹ for all sectors in 2010-2018).
North African emissions increased from 290 MtCO₂e yr⁻¹ in 1990-2000 to 305 MtCO₂eq yr⁻¹ in 2001-2009, and
further to 333 MtCO₂e yr⁻¹ in 2010-2018. Sub-Sahelian emissions decreased from 297 MtCO₂e yr⁻¹ in 1990-
2000 to 252 MtCO₂e yr⁻¹ in 2001-2009, and re-increased to 274 MtCO₂e yr⁻¹ in 2010-2018, a level smaller than
in the first decade (Fig. 6b). The Horn of Africa emissions increased from 149 MtCO₂e yr⁻¹ over 1990-2000, to
197 MtCO₂e yr⁻¹ over 2001-2009, and further to 260 MtCO₂e yr⁻¹ over 2010-2018. The emissions from Southern



Africa increased from 184 MtCO$_2$e yr$^{-1}$ in 1990-2000, to 180 MtCO$_2$e yr$^{-1}$ in 2001-2009, and further to 212
MtCO$_2$e yr$^{-1}$ in 2010-2018. Emissions from the Central African region increased from 111 MtCO$_2$e yr$^{-1}$ in 1990-
2000, to 114 MtCO$_2$e yr$^{-1}$ in 2001-2009, and further to 125 MtCO$_2$e yr$^{-1}$ in 2010-2018. We also computed the
GINI of African countries anthropogenic CH$_4$ per capita emissions and obtained the following values: 0.6 in
1990-1998, 0.5 in 1999-2008, 0.5 in 2009-2018, thus a trend of increasing 'inequality' between countries. As
compared to per capita FCO$_2$ emissions, more homogeneity is observed for CH$_4$ per capita emissions. Similar
to FCO$_2$ emissions, the GINI values remained stable over the three decades, showing a similar level of
inequalities over time.
**Bottom-up versus inversions for total and anthropogenic CH$_4$ emissions**

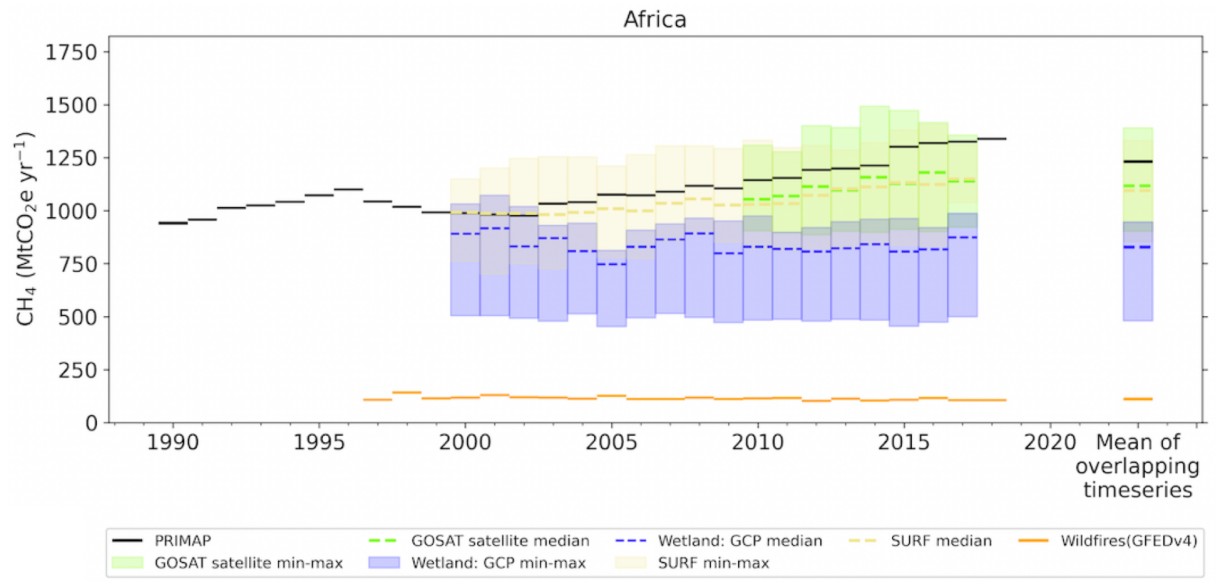

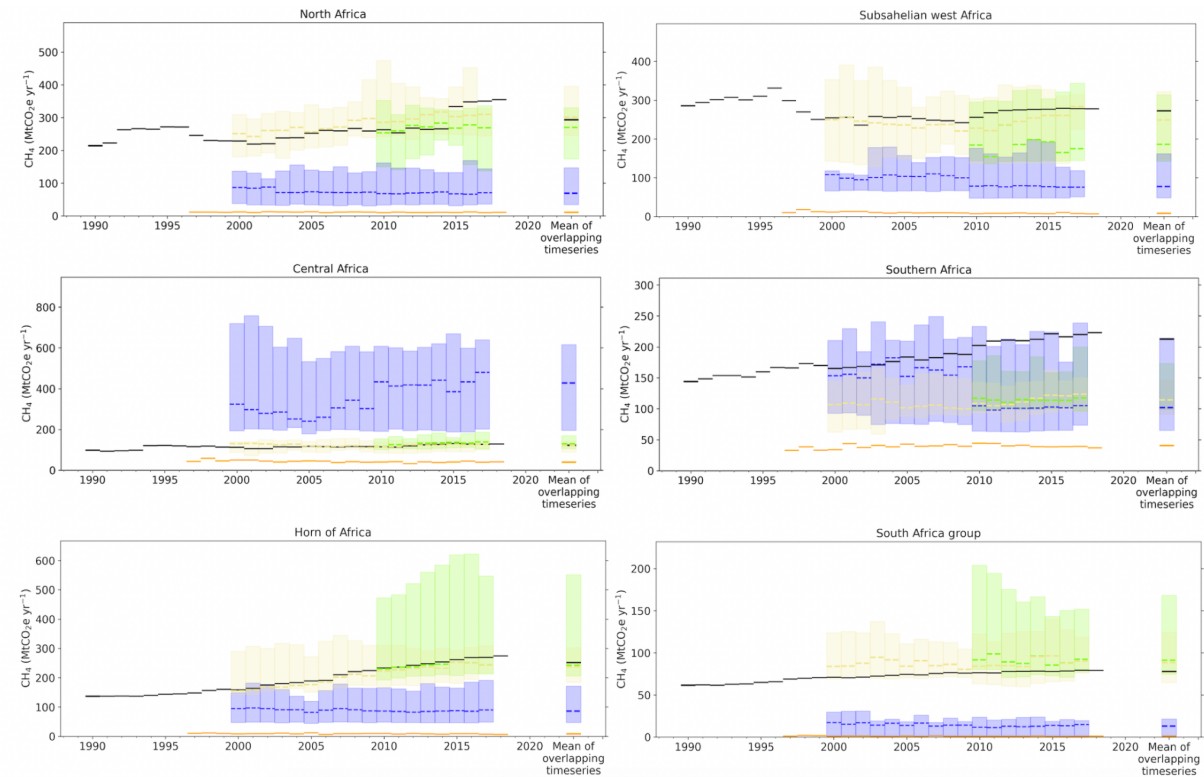

**Figure 7. Comparison of total anthropogenic CH$_4$ emissions in MtCO$_2$e yr$^{-1}$ from the PRIMAP-hist inventory (black) and global inversions. Shaded green and yellow areas represent the minimum and maximum range from GOSAT satellite and surface inversions, respectively. Shaded blue areas represent the minimum and maximum ranges of wetlands natural emissions from inversions. The orange lines represent wildfire emissions from GFED4.**

Figure 7 compares bottom-up anthropogenic emissions from PRIMAP-hist for the period 2000-2018 with inversions' anthropogenic emissions (see section 1). Wetlands natural emissions are shown in the figure only for information from the median and range of inversions. Over the overlapping time period, medians of both GOSAT and surface inversions are always smaller than PRIMAP-hist emissions, at continental and regional level, except for the Central African region. For the African continent, the mean and min-max of GOSAT inversions for anthropogenic CH$_4$ emissions over 2000-2018 is $1117^{1390}_{903}$ MtCO$_2$e yr$^{-1}$, very close to the mean of surface inversions of $1094^{1330}_{853}$ MtCO$_2$e yr$^{-1}$. A good agreement between GOSAT and surface inversions was also found in other high-emitting countries (Deng et al., 2021). In contrast, PRIMAP-hist gives a mean of CH$_4$ anthropogenic emissions of 1231 MtCO$_2$e yr$^{-1}$ over the period 2010-2017. The mean wetlands flux from inversions over 2010-2017 is of $827^{946}_{481}$ MtCO$_2$e yr$^{-1}$. Methane emissions from wildfires over Africa for the same period are less important, with a mean of 110 MtCO$_2$e yr$^{-1}$.





Regional emissions from PRIMAP-hist ranked in decreasing order are: North Africa (293 MtCO$_2$e yr$^{-1}$) > Sub-
Sahelian west Africa (272 MtCO$_2$e yr$^{-1}$) > Horn of Africa (252 MtCO$_2$e yr$^{-1}$) > Southern Africa (212 MtCO$_2$e
yr$^{-1}$) > Central Africa (123 MtCO$_2$e yr$^{-1}$) > South Africa (78 MtCO$_2$e yr$^{-1}$). For both GOSAT and surface
inversions, the ranking of regions (Table S10) is almost the same for surface inversions and PRIMAP-hist, with
the exception of Central Africa and Southern Africa.
**2.4 Results for N$_2$O emissions**

**N$_2$O PRIMAP-hist versus atmospheric inversions (total flux)**

**Total and sectoral N$_2$O anthropogenic emissions (PRIMAP-hist)**

**(a)**

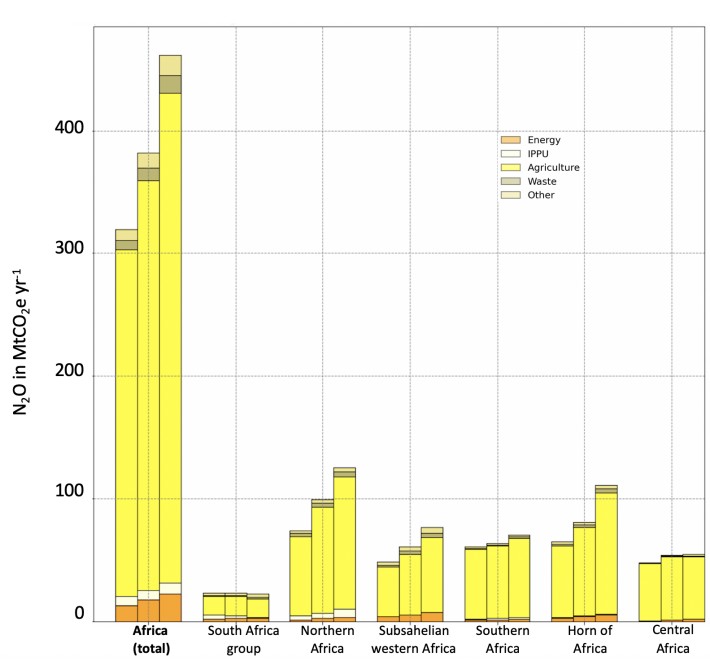



Data

**(b)**



**(c)**

**Figure 8. (a) African anthropogenic N₂O emissions in MtCO₂e yr⁻¹ over three decades: 1990-1998, 1999-2008 &**
**2009-2019. Data from PRIMAP-hist (2021). (b) Contribution of each sector to the change of African N₂O emissions**



**between the last three decades. (c) Same for different regions regrouping several countries. Data from PRIMAP-**
**hist (2021).**
Figure 8 presents anthropogenic $N_2O$ emissions from PRIMAP-hist, for five sectors (for country values, see
Fig. 4). Over the last three decades, the mean African emissions are 378 $MtCO_2e$ $yr^{-1}$, three times less than $CH_4$
emissions. The mean decadal $N_2O$ emissions increased from 319 $MtCO_2e$ $yr^{-1}$ in 1990-1999, to 382 $MtCO_2e$ $yr^{-1}$
in 2000-2009 (+20%), and further to 431 $MtCO_2e$ $yr^{-1}$ in 2010-2018. Over the last three decades, the main
emitting sector remained Agriculture. The $N_2O$ emissions increase also originates from Agriculture, with an
increase from 283 $MtCO_2e$ $yr^{-1}$ to 335 $MtCO_2e$ $yr^{-1}$ between 1990-1999 and 2000-2009, that is, +16.3 %
compared to of the total emission increase of +19.5%. The three other sectors show a smaller contribution to
the emissions increase: Energy (+1.4%), Other (+1%) and Waste (+0.8%). IPPU shows no change. Similarly,
between 2000-2009 and 2010-2019, the $N_2O$ emissions increase also came from the sector of Agriculture, with
an increase from 335 $MtCO_2e$ $yr^{-1}$ to 399 $MtCO_2e$ $yr^{-1}$ between 1990-1999 and 2000-2009.
The main contributing regions to the continental emissions are Northern Africa and the Horn of Africa (Fig.
8a). Between 2000-2009 and 2010-2019, the North African contribution increased from 99 $MtCO_2e$ $yr^{-1}$ to 125
$MtCO_2e$ $yr^{-1}$ (+27%). The main sectoral contribution is always Agriculture, which increased in that region from
86 $MtCO_2e$ $yr^{-1}$ to 107 $MtCO_2e$ $yr^{-1}$ (+21%). Emissions from the second largest emitting region, the Horn of
Africa, increased from 81.19 $MtCO_2e$ $yr^{-1}$ in 2000-2009 to 111 $MtCO_2e$ $yr^{-1}$ in 2010-2019 (+37%), mainly from
Agriculture. In the third most emitting region, Sub-Sahelian Africa, emissions increased from 61 $MtCO_2e$ $yr^{-1}$
in 2000-2009 to 77 $MtCO_2e$ $yr^{-1}$ in 2010-2019 (+27%), also from Agriculture. The least contributing region to
the increase of the total $N_2O$ emissions from 2000-2009 to 2010-2019 is South Africa which had a very small
decrease, mainly from IPPU (-6%) followed by Agriculture (-2%). On the contrary, there is a slight increase of
$N_2O$ emissions for the group of South Africa for the Other (+1%), Energy (+1%) and Waste (+1%) sectors.

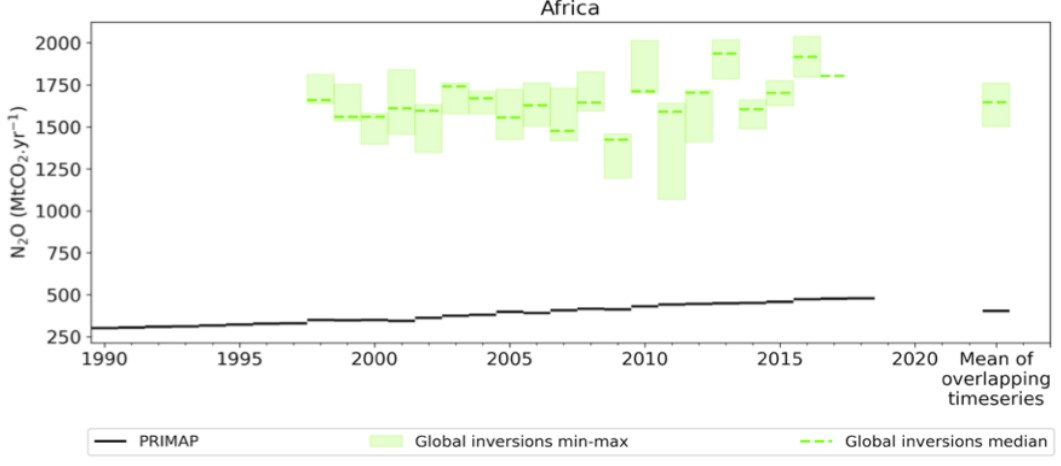

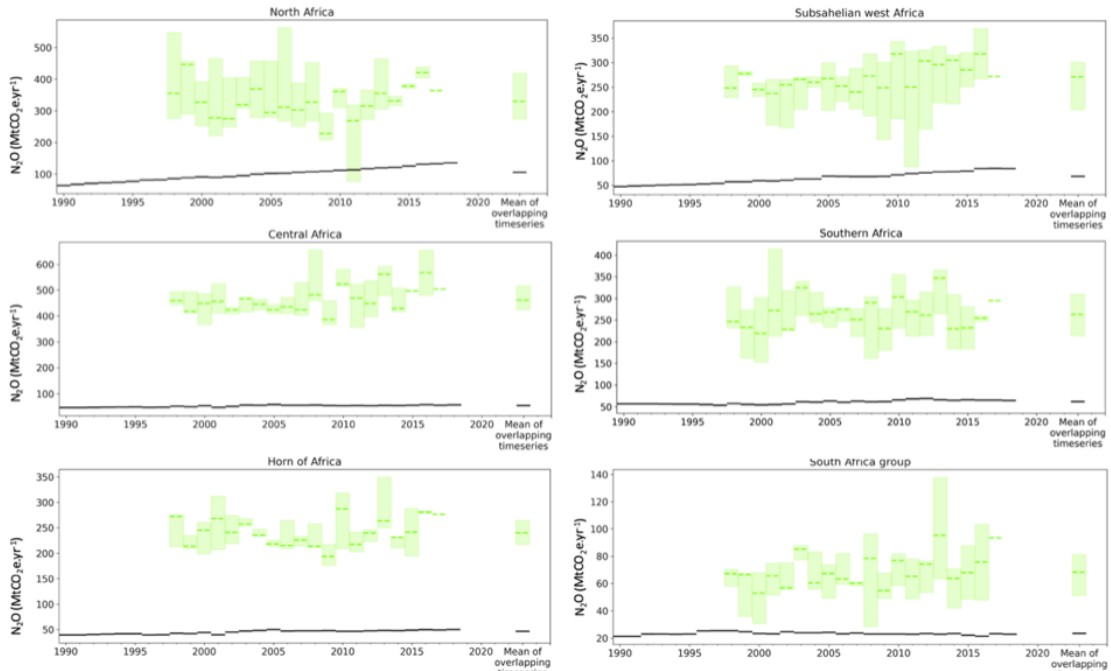

**Figure 9. Total N₂O emissions from PRIMAP-hist in MtCO₂e yr⁻¹ (black line) from three GCP atmospheric**
**inversions for the entire African continent and for six African sub-regions. The green line is the median of the three**
**inversions and the light green areas the maximum-minimum range.**
Figure 9 compares N$_2$O emissions from PRIMAP-hist and inversions. For total Africa, the mean of inversions
emissions over the overlapping time period 1998-2017 is $1647^{1760}_{1502}$ MtCO$_2$e yr$^{-1}$, much larger than the
PRIMAP-hist estimate of 360 MtCO$_2$e yr$^{-1}$. According to PRIMAP-hist, total African emissions increased by
28% between 1998 and 2017, while the trend of emissions from the inversions is 16 ± 8%. At regional scale,
emissions from inversions ranked in decreasing order are: Central Africa ($461^{517}_{424}$ MtCO$_2$e yr$^{-1}$) > North Africa
($330^{419}_{274}$MtCO$_2$e yr$^{-1}$) > Sub-Sahelian West Africa ($271^{330}_{68}$ MtCO$_2$e yr$^{-1}$) > Southern Africa ($263^{310}_{214}$MtCO$_2$e
yr$^{-1}$ > Horn of Africa $240^{265}_{217}$MtCO$_2$e yr$^{-1}$ > South Africa ($68^{81}_{51}$MtCO$_2$e yr$^{-1}$). According to PRIMAP-hist, the
ranking is: North Africa (106 MtCO$_2$e yr$^{-1}$) > Sub-Sahelian West Africa (68 MtCO$_2$e yr$^{-1}$) > Southern Africa
(62 MtCO$_2$e yr$^{-1}$) > Central Africa (54 MtCO$_2$e yr$^{-1}$) > the Horn of Africa (46 MtCO$_2$e yr$^{-1}$) > South Africa (24
MtCO$_2$e yr$^{-1}$) (See also Table S12). Emissions from PRIMAP-hist are smaller than inversions by a factor of 16.
This is likely due to the fact that we did not attempt to separate natural from anthropogenic emissions in
inversions. Other studies (Ciais et al., 2021; Petrescu et al., 2021 in Europe) showed that even after subtracting
N$_2$O natural estimates, inversions always point to higher estimates than BU methods.
**3 Discussion: synthesis for the three main GHG and comparison between BU and TD methods**



### 3.1 Synthesis of the steps for assessing net GHG trends over Africa

Here, we propose a first step towards the elaboration of what could become a more systematic method for a scientific benchmark of non-Annex I national inventories: 1) correct outliers, 2) a discussion about the realisms of estimates including considering geophysical aspects, 3) a proposal of an independent evaluation of inventory data by experts, 4) a comparison between UNFCCC data corrected thanks to expert judgment and other BU and TD methods, 5) computation of the mean of all BU and TD methods, 6) computation of "best fitted BU values" (meaning "best fitted BU values" excluding uncorrected UNFCCC data), and "TD values" (meaning "best fitted TD values": without considering $N_2O$ inversions replaced with PRIMAP-hist values), 7) identification of ranking anomalies.

### 3.2 Net GHG budget from bottom-up estimates

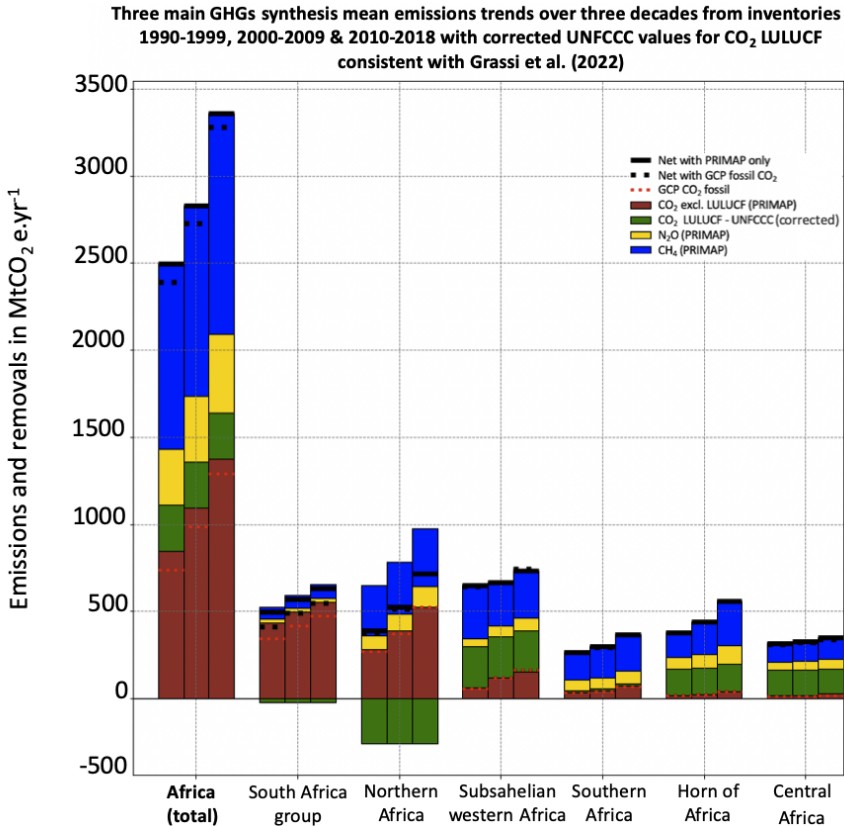

**Figure 10. Synthesis for the three main GHG from inventories (after UNFCCC LULUCF $CO_2$ corrections consistent with Grassi et al. (2022)) for the three main GHG with net African budget computation by BU inventories for Africa as a whole and for six sub-groups of African countries across three different decades (1990-1999, 2000-**





**2010, 2010-2018) using data and corrections from country inventories. Following the atmospheric convention,**
**positive numbers represent an emission to the atmosphere and the negative values represent a sink. Black**
**horizontal lines represent a net flux resulting from the addition of the three main GHG using PRIMAP-hist only,**
**dashed black horizontal lines also represent the net flux resulting from the addition of the three main GHG but**
**using the GCP dataset for FCO₂. Dashed red lines represent the fluxes from GCP FCO₂ available in the most recent**
**GCP paper, to compare them with PRIMAP-hist results which are represented with the brown bar plots. The N₂O**
**and CH₄ fluxes from PRIMAP-hist are respectively represented with yellow and blue bars. CO₂ emissions and**
**sinks from LULUCF are represented in green, they are taken from NC/BUR UNFCCC datasets with corrections**
**applied. Unit is MtCO₂e yr⁻¹.**
Figure 10 shows the budget for the three GHG from UNFCCC data with LULUCF data corrected using the
second approach. There is a clear increase of African total GHG emissions during the last 3 decades. The
differences between bottom-up datasets are mainly due to different sectoral allocations. However, the trends
are consistent and comparable, and differences among inventories tend to be less for the most recent decade.
**Table 5. Mean net total Africa and regional groups' emissions and removals from BU methods using either GCP**
**or PRIMAP-hist for FCO₂ over 2001-2017 in MtCO₂e.yr⁻¹.**

| | Type of dataset | | | | | | | | | |
|---|---|---|---|---|---|---|---|---|---|---|
| | **BU methods with GCP FCO₂** | | | | | | **BU methods with PRIMAP FCO₂** | | | |
| **Region** | | | | | | | | | | |
| | GCP + uncorrected UNFCCC LULUCF CO₂ | GCP + corrected UNFCCC LULUCF CO₂ as Grassi et al. (2022) | GCP + corrected UNFCCC LULUCF CO₂ as Grassi et al. (2022) but for DRC, NAM, NIG | GCP + median TRENDY v9 LULUCF CO₂ (min/max) | GCP + LULUCF CO₂ FAO total FL | | PRIMAP + uncorrected UNFCCC LULUCF CO₂ | PRIMAP + corrected UNFCCC LULUCF CO₂ as Grassi et al. (2022) | PRIMAP + corrected UNFCCC LULUC CO₂ as Grassi et al. (2022) but for DRC, NAM, NIG | PRIMAP + median TRENDY v9 LULUCF CO₂ (min/max) | PRIMAP + LULUCF CO₂ FAO total FL |
| **Africa total** | -599 | 2975 | 2122 | $2478^{4806}_{732}$ | 2728 | | -502 | 3069 | 2216 | $2572^{4899}_{827}$ | 2822 |
| **North Africa** | 613 | 589 | 589 | $835^{1216}_{549}$ | 839 | | 620 | 597 | 597 | $842^{1224}_{557}$ | 846 |
| **Central Africa** | -2605 | 316 | -448 | $-318^{633}_{-879}$ | 171 | | -2598 | 324 | -440 | $-310^{641}_{-871}$ | 179 |
| **Subsahelian West Africa** | 19 | 718 | 501 | $726^{1382}_{433}$ | 503 | | 15 | 714 | 497 | $723^{1378}_{430}$ | 500 |
| **Southern Africa** | 149 | 346 | 473 | $251^{953}_{-453}$ | 345 | | 151 | 347 | 475 | $252^{955}_{-452}$ | 346 |
| **South Africa group** | 640 | 524 | 524 | $542^{860}_{179}$ | 546 | | 719 | 603 | 603 | $621^{939}_{258}$ | 625 |
| **Horn of Africa** | 586 | 484 | 484 | $438^{805}_{-109}$ | 325 | | 587 | 484 | 484 | $439^{806}_{-108}$ | 326 |





692 At the country level, a small number of countries showed an increasing difference between PRIMAP-hist and

693 GCP estimates of fossil $CO_2$ emissions over time, but they are small $FCO_2$ emitters. The differences may also

694 be partly explained by changes in accounting methods as mentioned in Gütschow et al. (2016). The biggest

695 discrepancies are noticeable for Mali (64%), Cameroon (-62%), and the DRC (-38%), but those three countries

696 are not major $FCO_2$ emitters (Fig. 4.a-b).

697 Table 5 shows the differences of net African budget from various BU methods using GCP or PRIMAP-hist for

698 $FCO_2$ over 2001-2017 that are also illustrated on Fig. 11.

699 **Bottom-up LULUCF budget from UNFCCC corrected by Grassi**

700 Over 2001-2017 the net bottom-up GHG budget is 2975 $MtCO_2e$ $yr^{-1}$. Regionally the ranking in decreasing

701 order is: Sub-Sahelian West Africa (718 $MtCO_2e$ $yr^{-1}$) > North Africa (588 $MtCO_2e$ $yr^{-1}$) > South Africa group

702 (524 $MtCO_2e$ $yr^{-1}$) > Horn of Africa (484 $MtCO_2e$ $yr^{-1}$) > Southern Africa (346 $MtCO_2e$ $yr^{-1}$) > Central Africa

703 (316 $MtCO_2e$ $yr^{-1}$).

704 **Bottom-up LULUCF budget $CO_2$ from FAO**

705 The bottom-up budget from FAO data is 2728 $MtCO_2e$ $yr^{-1}$, 8% less than above. The ranking of regions in

706 decreasing order is: North Africa (838 $MtCO_2e$ $yr^{-1}$) > South Africa group (546 $MtCO_2e$ $yr^{-1}$) > Sub-Sahelian

707 West Africa (503 $MtCO_2e$ $yr^{-1}$) > Southern Africa (345 $MtCO_2e$ $yr^{-1}$) > Horn of Africa (325 $MtCO_2e$ $yr^{-1}$) >

708 Central Africa (171 $MtCO_2e$ $yr^{-1}$).

709 **Bottom-up LULUCF budget from DGVMs**

710 The net GHG budget for Africa is of 2478 $^{4806}_{733}MtCO_2e$ $yr^{-1}$ $MtCO_2e$ $yr^{-1}$, 9% less than with FAO. The ranking

711 of regions in decreasing order is: North Africa (835 $^{1216}_{549}$ $MtCO_2e$ $yr^{-1}$) > Sub-Sahelian West Africa

712 (726 $^{1382}_{433}MtCO_2e$ $yr^{-1}$) > South Africa (542 $^{859}_{179}$ $MtCO_2e$ $yr^{-1}$) > Horn of Africa (438$^{805}_{-109}$ $MtCO_2e$ $yr^{-1}$) >

713 Southern Africa (251 $^{953}_{-453}MtCO_2e$ $yr^{-1}$ > Central Africa ($-318$ $^{633}_{-879}$ $MtCO_2e$ $yr^{-1}$).





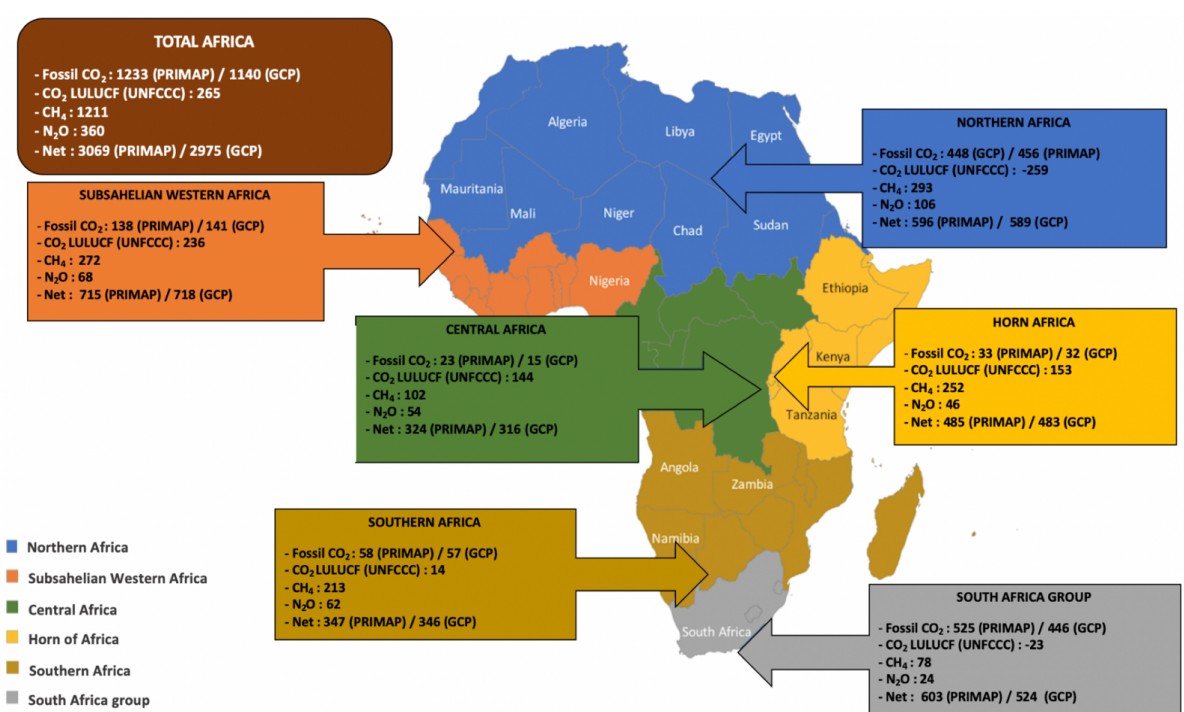

**Figure 11. 2001-2018 emissions in MtCO₂e yr⁻¹ for fossil CO₂ (GCP and PRIMAP-hist), LULUCF CO₂ (corrected UNFCCC data consistent with Grassi et al. (2022), CH₄ (PRIMAP-hist), N₂O (PRIMAP-hist) for Africa, and for six regions.**

### 3.3 Net GHG budget from inversions

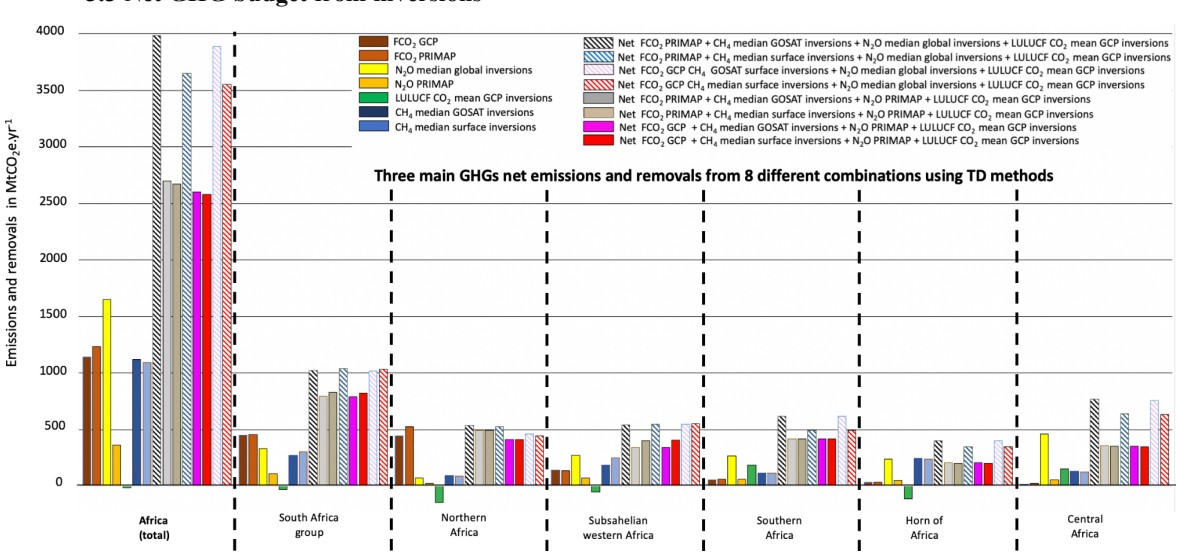



**Figure 12. Synthesis for the three main GHG with net African budget computation by all TD methods for Africa**
**as a whole and for six sub-groups of African countries across overlapping time series (2001-2017). Following the**
**atmospheric convention, positive numbers represent an emission to the atmosphere and the negative values**
**represent a sink. The CO₂ emissions and sinks from LULUCF are represented in green, they are taken from GCP**
**2020 dataset. Unit is MtCO₂e yr⁻¹.**
Figure 12 shows different combinations of inversion GHG budgets and individual gasses contributions.
For total Africa, the mean net GHG budget from inversions where $N_2O$ inversions are replaced by PRIMZP-
hist is 2638 $^{5873}_{1761}$ MtCO₂e yr⁻¹, differing only by 1 % from the bottom up GHG budget. Regional GHG budgets
in decreasing order are: North Africa (810 $^{1170}_{279}$ MtCO₂ yr⁻¹) > South Africa group (452 $^{751}_{161}$ MtCO₂ yr⁻¹) >
Southern Africa (416$^{1465}_{-334}$ MtCO₂ yr⁻¹) > Sub-Sahelian West Africa (373$^{1051}_{36}$ MtCO₂ yr⁻¹) > Central Africa
(352$^{1592}_{-1133}$ MtCO₂ yr⁻¹) > Horn of Africa (204$^{873}_{-456}$ MtCO₂ yr⁻¹) (Table S16). The mean net of inversions
including $N_2O$ inversions is substantially higher, 3879 $^{7341}_{1320}$ MtCO₂e yr⁻¹. Regional GHG budgets in decreasing
order are: North Africa (1034 $^{1475}_{600}$ MtCO₂e yr⁻¹) > Central Africa (759 $^{2054}_{-763}$ MtCO₂e yr⁻¹) > Southern Africa
(616 $^{1713}_{-262}$MtCO₂e yr⁻¹) > Sub-Sahelian West Africa (576 $^{1313}_{-61}$MtCO₂e yr⁻¹) > South Africa group
(496 $^{814}_{138}$ MtCO₂e yr⁻¹) (Table S16).
### 3.4 Comparison between bottom-up and top-down methods

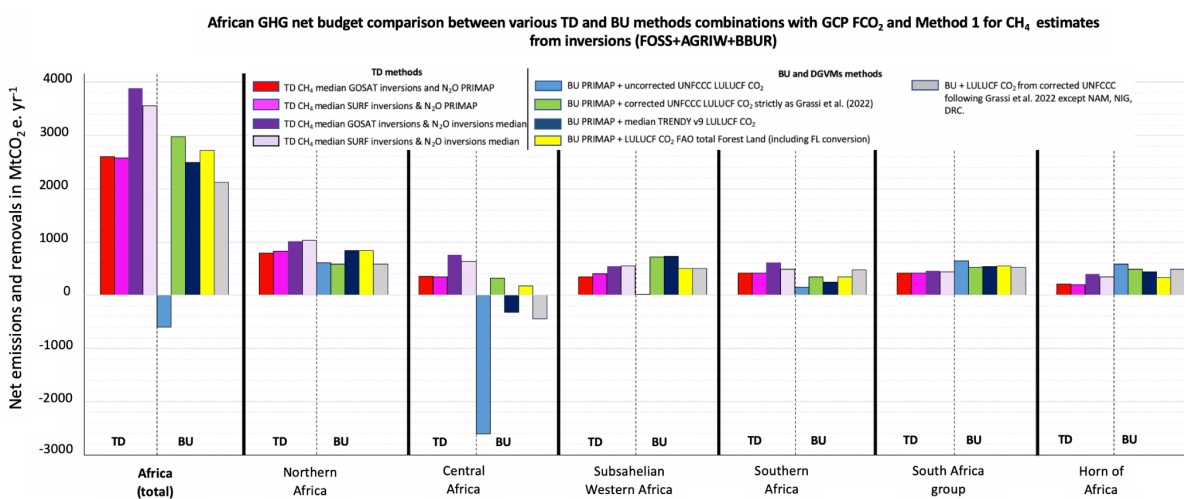

**Figure 13. Synthesis for the three main GHG net African budget from TD and BU methods. using Method 1**
**regarding anthropogenic CH₄ estimated from inversions (FOSS+AGRIW+BBUR) for comparative net emissions**
**and removals computation by BU and TD methods for Africa as a whole and for six sub-groups of African countries**
**across the overlapping period (2001-2017). FCO₂ data from GCP. N₂O from global inversions and from PRIMAP-**
**hist. For TD methods, anthropogenic CH₄ from both GOSAT and surface inversions are used, and LULUCF from**





**GCP inversions only. For BU methods, anthropogenic CH$_4$ and N$_2$O from PRIMAP are used, and with five different**
**methods for assessing LULUCF CO$_2$: from uncorrected UNFCCC data; from corrected UNFCCC data according**
**Grassi et al. (2022); from corrected UNFCCC except Namibia, Nigeria and DRC; from TRENDY v9; from FAO**
**FL including FL conversions. Following the atmospheric convention, positive numbers represent an emission to**
**the atmosphere and the negative values represent a sink. All values are in MtCO$_2$e.**
Figure 13 shows the GHG budgets from all combinations of bottom-up and top-down methods. The mean of
all methods after filtering outliers (Grassi et al. (2022) UNFCCC corrections, using PRIMAP instead of
inversions for N$_2$O) is $2630\ ^{4557}_{1974}$ MtCO$_2$e yr$^{-1}$, which represents only % of global FCO$_2$ emissions. The mean
of all estimates points out to a source in the six African regions ranked in decreasing order as: North Africa
($761\ ^{988}_{460}$ MtCO$_2$e yr$^{-1}$) > South Africa group ($513\ ^{702}_{161}$ MtCO$_2$e yr$^{-1}$) > Horn of Africa ($318\ ^{699}_{-80}$ MtCO$_2$e yr$^{-1}$) >
Sub-Sahelian West Africa ($492\ ^{913}_{286}$ MtCO$_2$e yr$^{-1}$) > Southern Africa ($354^{998}_{-78}$ MtCO$_2$e  yr$^{-1}$) > Central Africa
($143\ ^{882}_{-670}$ MtCO$_2$e yr$^{-1}$).
**3.5 Uncertainties specific to DGVM / inversions for LULUCF CO$_2$**
In Fig. 5, we showed important disagreements among models regarding LULUCF CO$_2$ on whether Africa has
been a small source over the last 20 years (as shown by inversions) or a net sink (as shown by DGVM and
UNFCCC except with the Grassi et al. correction). There is also more interannual variability in the DGVM
results, mainly from climate, which is absent from UNFCCC as inventories provide only decadal smoothed
flux estimates. The larger sink in the DGVM compared to the corrected UNFCCC estimates using the method
of Grassi et al. (2022) may be due to the fact that non-Annex I UNFCCC estimates generally do not include
dead biomass or harvested wood products. If forest biomass is estimated by a stock-change approach,
therefore, changes in living biomass due to transfer to dead biomass and harvested wood products will be
considered emitted in that year, while in the DGVM it will decay more slowly over time. Another difference
is the treatment of land use change emissions, based on historical global land use change maps for the DGVM,
which can significantly differ from national land use datasets. On the other hand, DGVM do not represent
forestry and may underestimate sinks in intensively managed young forests. Finally, DGVM do not separate
between unmanaged and managed lands, while UNFCCC inventories only account for managed land, yet
including conservation areas and indigenous territories. Grassi et al. (2022) showed that the difference
between the global UNFCCC sink (1100 MtCO$_2$ yr$^{-1}$) and the global land carbon sink (4767 MtCO$_2$e yr$^{-1}$)
must be explained by the contribution of non-managed lands. But in the case of Africa, it was not possible to
extract from UNFCCC reports the national areas of unmanaged land, and we had to also look at UNFCCC
Technical Assessment Reports (TAR) as well as REDD+ reports to extract information. Methods of
assessment have not been fully standardized since 1990, and they differ depending on the countries analyzed,





and on the emissions categories considered. In this context, when comparing UNFCCC estimates with data
from DGVM and inversion models, different layers of aggregated uncertainties affect the analysis. (Deng et
al., 2021; Petrescu et al., 2021; Grassi et al., 2018).

### 3.6 Differences between bottom-up and top-down $CH_4$ emissions

The methodology used for removing natural $CH_4$ emissions from inversions is key for comparing with bottom-
up estimates. In this paper, we used a separation based on the natural emissions solved by each inversion
(section 2.3 method 1). Using an alternative method from Deng et al. (2022) based on natural emissions from
the median of all inversions gives smaller anthropogenic emissions than PRIMAP-hist (Fig. S10).

### 3.7 Differences between bottom-up and top-down $N_2O$ emissions

For $N_2O$ emissions, discrepancies between inventories and inversions are very high, especially for the group of
Central African countries, where the vegetation covers an important land area with likely large natural $N_2O$
(Deng et al., 2022). We can suppose that more broadly for all African groups, the lack of accounting of natural
emissions is the main reason why PRIMAP-hist estimates are much smaller than inversions. All African
countries used Tier 1 emission factors and include only direct $N_2O$ emissions. The study by Deng et al. (2022)
underlined that indirect anthropogenic emissions notably coming from "atmospheric nitrogen deposition and
leaching from anthropogenic nitrogen additions to aquifers and inland water are usually not reported by non-
Annex I countries" and that this under-reported source of anthropogenic emissions tends to represent about 5%
to 10% of anthropogenic $N_2O$.

### 4 Summary, concluding remarks and perspectives

Africa is a large continent gathering 56 countries, and some countries are major GHG emitters. Because of its
rapidly growing population and high industrial potential, Africa is a critical geography regarding climate
change mitigation and adaptation policy. Depending on the emissions pathways, Africa, which is already a big
emitting region, is expected to represent between at least a bit more than 10% of the global share by 2050, and
could become as high as 18% of global emissions by 2050 (van der Zwaan, 2018).
This paper delivers both a continental view and a detailed analysis of the three main GHG trends during the
last thirty years across this continent as a whole, across relevant groups of countries given the inversions'
resolutions, and also considering country details. Thanks to the comparison of different methods and datasets,
the uncertainty about the net emissions and removals of GHG lowers. The interest of studying Africa is high
not only from a scientific point of view, but also from a climate-policy perspective, as under the UNFCCC



principle of "common but differentiated responsibility" about global warming, the credibility of the PA lies in
the effective participation and inclusiveness of all parties, including non-Annex I countries. Our effort of
comparing BU datasets and inversions and analyzing differences for African GHG emissions and removals
assessment by looking at trends since 1990 will also be useful for future updates on a regular basis within the
2023 GST perspective.
At the scale of Africa, there is a rapid increase of $FCO_2$ emissions that roughly doubled since 1990. This increase
is dominated by coal emissions for the decade 1990-1998 compared to 1999-2008 (+9%), and by oil for the
decade 1999-2008 compared to the decade 2008-2017 (+16%). As for $CO_2$ LULUCF, we found that BU
estimates are featured with important annual fluctuations, as opposed to periodic national inventories
assessments, the reconciliation between the sectoral classification for anthropogenic estimates between TD and
BU has to be done "manually" and is not uniform to date, which doesn't facilitate the comparability of those
different approaches. There are also differences among GCP inversions for $CO_2$, due to the fact that choices of
model transport may differ among models, because prior fluxes can also differ between modeling teams, and
because the African GHG observation network is featured with few stations and relatively scarce data. The lack
of integration of $CO_2$ lateral anthropogenic and river fluxes is also an issue to be taken into account when trying
to compare BU and TD methods (Ciais et al., 2022), and in the present study we did integrate those lateral
fluxes. Anthropogenic $CH_4$ from PRIMAP-hist estimates indicate that out of the total African emissions
increase from 1064 $MtCO_2e$ $yr^{-1}$ to 1116 $MtCO_2e$ $yr^{-1}$ between 1990-2000 and 2001-2009 (+5%), only two
sectors contributed: Agriculture, in a dominant way (+8%) and Waste (+5%). Energy contributes to emissions
decrease (-8%) that is however too small to offset other sectors' $CH_4$ emissions that represent a net increase.
The main regional contributions come from North Africa and from the Agriculture sector (+12%). Over the
same period, the least contributing emitter is the group of South Africa (+12%), with only one decreasing
emissions sector: Agriculture (-1%). The mean 2001-2009 emissions increased by +15% over 2010-2018 with
contribution from all sectors except IPPU. This increase is dominated by Agriculture (+8%) and Waste (+ 6%).
For 2010-2018, the two main contributing regions for $CH_4$ emissions are Northern Africa and Sub-Sahelian
Western Africa, Agriculture being the dominant emitting sector. From inversions, after withdrawing natural
emissions and wildfires using the GFED dataset from total $CH_4$ emissions, median values are almost always
below PRIMAP-hist estimates. $CH_4$ natural emissions have an important impact in Africa especially in the
Central African region as well as in the Southern countries. $N_2O$ TD estimates are always higher than the ones
from PRIMAP-hist, underlining the importance to separate natural $N_2O$ emissions from total estimates in order
to deliver appropriate anthropogenic assessments thanks to the inversions.
To compute a net budget for the three main GHG emissions and removals and for comparability we used the
$MtCO_2e$ $yr^{-1}$ metric and latest IPCC report recommended GWP. The choice of a constructed GWP metric,
however, creates additional associated uncertainties notably due to the selected time horizon. By computing



the mean of methods excluding uncorrected UNFCCC and $N_2O$ inversions data from twenty different ways for
assessing GHG emissions and removals in Africa, we found that the most recent net from the three main GHG
in Africa is a source of $2630^{4557}_{1974}$ $MtCO_2e$ $yr^{-1}$.
Our assessment of African GHG emissions trends over 30 years through different methods can enable
comparisons of *ex post* with *ex ante* pledges of the PA, whose baseline year is often 1990. However, given the
global geopolitics to date featured with the prevailing principle of national sovereignty, a scientific assessment
of GHG can only work as a supporting tool (Janssens-Maenhout et al., 2020) and cannot be directly policy-
prescriptive. We note a relatively good match among the various types of estimates in terms of overall trends,
especially at a regional level and on a decadal basis, but large differences even among similar typologies of the
methods (TD or BU). The large discrepancies are a scientific limit to the possibility of precise verification of
the African country-reported emissions, but they are good enough to indicate trends. To compute a net from
the three main GHG, no purely "TD" method is available due to the necessity to replace $N_2O$ inversions data
with BU data. An original result of this study is that we proposed at a small scale what may become a systematic
formalized methodological protocol for independent verification of a net estimate using country-reported data,
to be possibly implemented at the UNFCCC secretariat scale in a centralized way. The African GHG increasing
trend is not in line with the mitigation aims of the PA towards net-zero globally. Research teams focusing on
inversion methods (Nickless et al., 2020), underline that uncertainties should not be above 15% in order to
deliver a reasonable verification support capacity. A major source of complexity for the evaluation of the
respect of the Paris Agreement comes from the fact that national pledges generally fall below the discrepancies
between different scientific independent estimates. This calls for investments not only in improvements of
atmospheric measurement devices but also in the research efforts for standardizing verification methods. At
the policy level, the extrapolation of this study to the climate policy field could also serve as a compelling
argument for the creation of a global dedicated "Climate Inspection task force" of the UNFCCC.
**5 Data availability**
The datasets from the three main greenhouse gasses used in this paper ($CO_2$, $CH_4$, $N_2O$) from the various BU
inventories, TD inversions and DGVM over Africa will be made publicly available. This database is available
from Zenodo at: https://doi.org/10.5281/zenodo.7347077 (Mostefaoui et al., 2022).
This dataset contains 32 data files:
**- $CO_2$ inversions** (annual flux for LULUCF $CO_2$)
- African $CO_2$ TD inversions GCB2020 1990-2019: annual $CO_2$ flux from GCB inversion models
- African $CO_2$ lateral flux 2001-2019: annual $CO_2$ lateral flux including river transport, crop and wood
product trade.
- African $CO_2$ TRENDYv9 1990-2019: annual $CO_2$ flux from 14 DGVM





- FAO 1990-2019: annual emissions and removals from FAO dataset

- Inventory IPCC 1990-2019: annual flux from inventory data collected from UNFCCC national

inventories in the IPCC categories

**- CH₄ inversions 2000-2017** (annual flux)

- African CH₄ global inversion 2000-2017: CH4 flux over 2000-2017 from 11 in situ inversion and 11

satellite inversion models from four sectors; fossil refers to emissions from the fossil sector; agriculture and
waste refers to emissions from both the agriculture and waste sector; biomass burning refers to emissions from
biomass burning

- GFEDv4 1997-2016: wildfire emissions from the Global Fires Emission Dataset (GFED) version 4

**- N₂O inversions 1998-2017** (annual flux)
- N₂O PYVAR 1998-2017: total N₂O emissions from PyVAR inversions;
- N₂O TOMCAT-INVICAT 1998-2015: total N₂O emissions from TOMCAT-INVICAT model;
- N₂O MIROC4 - ACTM  1998-2016: total N₂O emissions from MIROC4-ACTM model;
Data used in this study are also included in the Supplementary Information (for example, from FAO data) and
on public websites (CDIAC, PRIMAP-hist, World Bank data). Any other data that support the findings of this
study are available from the corresponding author upon request.
**Author contributions.** MM, PC, PP and MJM designed research and led the discussions; MM wrote the initial
draft of the paper and edited all the following versions; MM made all figures ; MJM and PP processed the
original data from inversions and DGVM; MM processed the UNFCCC data and corrections; PC, PP and YE
gave valuable suggestions to the manuscript structure; PC, MJM and PPP read, gave comments and advice on
previous versions of the manuscript; all co-authors commented on specific parts related to their datasets; PC,
MJM, PP, FC, SS, CR, IL, MS, PP are data providers.
**Competing interests.** The authors declare that they have no conflict of interest.
**Disclaimer.** The views expressed in this publication are those of the authors.
**Acknowledgements**
MM acknowledges funding from Sorbonne University, Institute of the Environmental Transition. PC, PP, and
MJM were supported by the European Commission, Horizon 2020 Framework Program (VERIFY, grant no.
776810). PC and YE are also supported by the RECCAP2 project (grant no. ESRIN/4000123002/18/I-NB).
We acknowledge Stephen Sitch and Trendy modelers for the use of their dataset. We also acknowledge
Christian Rödenbeck for the use of CarboScope, Frédéric Chevallier for CAMS, Ingrid Luijkx for CTE,





Marielle Saunois for $CH_4$ inversions. The PyVAR-N2O modeling results were provided by Rona Thompson
(NILU)    and    were    funded    through    the    Copernicus    Atmosphere    Monitoring    Service
(https://atmosphere.copernicus.eu/), implemented by ECMWF on behalf of the European Commission, and
were generated using computing resources from LSCE.

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
