# Peer review of "Greenhouse gasses emissions and their trends over the last three decades across"

_Earth System Science Data, 2023_

## Author Comment (AC1)

**REPLY TO REFEREE #2 CHRIS JONES**

We warmly thank Chris Jones for his very thoughtful and precious Referee comments and for the fact that the Referee acknowledges the manuscript content.

Below we provide answers to the comments posted by Chris Jones.

"I have a few minor comments which may help presentation, but recommend publication once these are addressed."

Thank you, we will ensure that all the comments are addressed.

"My main comment is that due to its length, I think there may be a cleaner way to summarize and finish the paper. When I got to figure 10 (and also fig 11) I thought "this is a nice synthesis plot" (i.e. it feels like a really good place to finish and summarize everything you've found), but then the paper goes back into more details and uncertainties – I was then left not knowing whether to believe figures 10 and 11 any more…. It would be nice maybe to have the discussion of uncertainties first, and then finish with a clear synthesis figure like 10/11 which shows your final best estimates."

We agree that the paper should be shortened after figures 10 and 11 and will ensure that the discussion about uncertainties is before these figures.

**Minor comments**

- "You note that land use $CO_2$ emissions have the greatest uncertainty. It is worth noting that this is true globally and in fact land-use $CO_2$ are the most uncertain of all GHGs (at least when expressed as $CO_2e$) according to IPCC."

We will reframe that point and refer to IPCC reports.

- "For $N_2O$, you note the difference between bottom-up and top-down estimates. I'm curious how this compares with the global situation – is this similarly affected by the potential contribution of natural sources? Or is this just a problem for Africa (and if so why?)."

According to Deng et al. (2022), the global situation from inversions for main emitters is similarly affected by the potential contribution of natural sources as well, which is difficult to estimate / separate.

We copied below Fig. 11 from Deng et al. (2022) showing that even when removing 'intact / non managed lands' from inversions, in many countries, especially tropical countries, the inversions give a systematically much higher anthropogenic emission of $N_2O$ than inventories, suggesting that there are

either missing anthropogenic sources or some 'natural' sources (e.g. conservation areas) in managed lands being underestimated by inventories. We will highlight this point in the revised manuscript.

[Figure]

Figure 11 from Deng et al. ESSD 2022

- "Given the importance of this uncertainty – could the paper recommend which estimate to use? Do you believe the top-down or bottom-up ones are better? At the moment it just leaves the reader to see two very different possible answers…."

Thanks, it is a tricky point, we will add a more detailed paragraph about it based on our reply to the comment above.

- "Table 1: why do you list all 3 $N_2O$ inversions as different lines here, but for $CO_2$ you just have "GCB ensemble" (which is itself multiple models)?"

We will suppress the listing of all 3 $N_2O$ inversions and refer them to the $N_2O$ budget paper.

- "Line 291 – Pongratz is a good reference here for the issues of "loss of sink capacity". https://esd.copernicus.org/articles/5/177/2014/esd-5-177-2014.pdf"

Thanks, we will add reference to Pongratz and her mathematical definition of loss of additional sink capacity" (LASC) as detailed p. 180 of her paper as:

"LASC $=\delta(Em - Ep)^3$

"$CO_2$ fluxes in response to environmental changes on managed land as compared to potential natural vegetation. Historically, the potential natural vegetation would have provided a foregone sink as compared to human land use."

- "Line 339 – why do you use GWP100 numbers from AR4 (16 years old?) – can you update these to AR6?"

We used AR4 because many African countries have been following 2006 IPCC guidelines referring to AR4 GWP. 2019 refinement to IPCC guidelines do not recommend any specific metrics, therefore we are not following IPCC guidelines used by countries. We have explained this point and given the coefficients to use to change AR4 to AR6 GWP values in the revised manuscript.

---

## Author Comment (AC2)

**REPLY TO THE ANONYMOUS REFEREE #1**

We thank Referee #1 for the valuable comments in the interactive discussion of our ESSD preprint review article and for acknowledging the "crucial" interest of this study.

"The paper provides an analysis of greenhouse gas (GHG) emissions and trends in Africa over the past three decades, focusing on evaluating different datasets and their potential for verifying official country-reported data. The study examines emissions of carbon dioxide (CO2), methane (CH4), and nitrous oxide (N2O) using both bottom-up approaches (such as national inventories and ecosystem models) and top-down methods (including atmospheric inversions).

The findings contribute to understanding emission trends and uncertainties in Africa, which is crucial for climate policy and the goals of the Paris Agreement. Overall, the topic is interesting.

However, I have some concerns as follows:

1. In addition to providing the datasets (https://doi.org/10.5281/zenodo.7347077), this paper needs to include datasets usage (quality control method, datasets limitation, etc.)"

We agree and will add a detailed section named datasets usage including quality control method, datasets limitation in the Zenodo repository.

2. "The method for calculating trends needs to be described and the impact of different trend calculation methods on trend results needs to be discussed."

Thanks, the revised paper will contain the description of the different computation methods on trends with further details. (For estimating linear trends and their significance, we used the R Python function to compute the correlation coefficient for medians values over overlapping time periods, that we will describe more in detail. We also computed GINI for emissions per GDP that we will further detail).

3. "As statistics play a crucial role in this study, it is important to provide further details, such as confidence intervals, to ensure its robustness."

Thanks for your comment, we agree with Referee#1 that confidence intervals are critical. Given that some of our estimates are based on a small number of models / estimates, we cannot calculate the full distribution and a 95% CI but we rather reported ranges with min / max. Assuming that the unknown distributions would be Gaussian, like in Schultze et al. (2011) we could infer a 2-sigma ($\approx$ 95%) CI if we assumed that min-max are equivalent to 3-sigma, but in view of the small numbers of estimates e.g. for $N_2O$ with only 3 inversions, we prefer to just give the min-max range. Moreover, for NGHGI, this is more tricky and as

all African countries are non-Annex I, they unfortunately do not deliver confidence intervals but Grassi et al. (2022) estimated for $CO_2$ LULUCF fluxes uncertainties of 50 % for the average of non-Annex-1 countries, which we mentioned in the text and used by default in the revised manuscript.

We extended the discussion on uncertainties in section 1 (methods and datasets), we added the following paragraph about the underlying data uncertainty description in the method section (page 6, lines 173-180):

> *"No specific standard guidelines currently exist for defining uncertainties for datasets from BU and TD data products. In general, uncertainty estimates are understood as the spread among minimum and maximum values from one methodology. A main source of uncertainty in the comparison of country-reported data with other data products is the inclusion or not of natural fluxes additionally to anthropogenic emissions sectors. For inversions, the prior geospatial distribution of emissions is a critical source of uncertainty. For the comparability of the different data products presented in this study, we discuss only the mean value over the period of overlapping data availability. Referenced datasets are available at https://doi.org/10.5281/zenodo.7347077(Mostefaoui et al., 2022)."*

In the discussion paragraph 3.5 about uncertainties for DGVM and inversions for LULUCF $CO_2$ (pages 41-42, lines 751-773) we also reminded how uncertainties were defined for each method while discussing "unknown-unknown' types of uncertainties.

4. "When employing in situ surface networks for dataset validation, are there specific factors, such as latitude, longitude, climate zones, etc., that exhibit correlations with the product's quality?"

Thanks for the question. May we please ask to what line of the paper exactly does the anonymous referee #1 refer to? We have not used in situ for dataset validation per se, only the GOSAT data were evaluated against TCCON independent ground based total column $XCH_4$.

The African ground-based network is very sparse. There are only three currently active surface flasks over this whole continent, located in Namibia (Gobabeb), in the Seychelles (Mahe Island), and in South Africa (Cape Point). The one in Algeria (Assekrem) was terminated on 26/08/2020, and the one in Kenya has been inactive since 21/06/2011. We summarize the characteristics of the surface flasks in Africa, available on the NOAA website in the table below:

| Station name, Country | Parameter | First sample date | Status for the three GHG | Frequency | Elevation (in meters above mean sea level) | Cooperating Agencies |
|---|---|---|---|---|---|---|
| Assekrem, Algeria | $CO_2$
 $CH_4$
 $N_2O$ | 12/09/1995
 12/09/1995
 12/09/1995 | Terminated since 26/08/2020 | Discrete Monthly | 2710 | Algerian National Office of Meteorology |
| Gobabeb, Namibia | $CO_2$
 $CH_4$
 $N_2O$ | 13/01/1997
 13/01/1997
 13/01/1997 | Ongoing | Discrete Monthly | 456 | Gobabeb Training and Research Center |
| Mahe Island, Seychelles | $CO_2$
 $CH_4$
 $N_2O$ | 15/01/1980
 12/05/1983
 13/06/1997 | Ongoing | Discrete Monthly | 2 | Seychelles Bureau of Standards |
| Cape Point, South Africa | $CO_2$
 $CH_4$
 $N_2O$ | 5/01/1980
 12/05/1983
 13/06/1997 | Ongoing | Discrete Monthly | 230 | South African Weather Service |
| Mt. Kenya, Kenya | $CO_2$
 $CH_4$
 $N_2O$ | 11/02/2010
 11/02/2010
 11/02/2010 | Inactive since 21/06/2011 | Discrete Monthly | 3644 | Kenya Meteorological Department |

**Table with Surface flasks characteristics over the African continent. Data synthetized from NOAA website.**

5. "Lines 89-90. BU methods and TD methods need to be further explained."

Thank you, we will add more detailed explanations in the revised manuscript.

6. "Line 124 and Line 162. Redefinition for the abbreviation "bottom-up (BU)". You have already defined it in line 89. Please recheck the manuscript to ensure that the same error does not occur."

Thanks, we will make sure that acronyms / abbreviations are only defined once.

---

## Author Response (AR2)

**REPLY TO CHRIS JONES**

We warmly thank Chris Jones for his very thoughtful and precious Referee comments and for the fact that the Referee acknowledges the manuscript content.

Below, in black color, we provide answers to the comments posted by Chris Jones (in blue).

"I have a few minor comments which may help the presentation, but recommend publication once these are addressed."

Thank you, we ensured that all the comments were addressed.

"My main comment is that due to its length, I think there may be a cleaner way to summarize and finish the paper. When I got to Fig.10 (and also Fig. 11) I thought "this is a nice synthesis plot" (i.e. it feels like a really good place to finish and summarize everything you've found), but then the paper goes back into more details and uncertainties – I was then left not knowing whether to believe figures 10 and 11 any more… It would be nice maybe to have the discussion of uncertainties first, and then finish with a clear synthesis figure like 10/11 which shows your final best estimates."

We agree that the paper should be shortened after Fig. 10 and Fig.11 (numbered as such in the previous version of the manuscript) and we ensured that those figures are now concluding figures as requested: they are now named Fig. 12 (line 977) and Fig. 13 (line 1022) of the revised version. We ensured that

the discussion about uncertainties is before these figures see section 3.1, section 3.2 and section 3.3 (lines 851 to 902).

**Minor comments**

- "You note that land use $CO_2$ emissions have the greatest uncertainty. It is worth noting that this is true globally and in fact land-use $CO_2$ are the most uncertain of all GHGs (at least when expressed as $CO_2e$) according to IPCC"

Thank, we underlined that point lines 874 to 875.

- "For $N_2O$, you note the difference between bottom-up and top-down estimates. I'm curious how this compares with the global situation – is this similarly affected by the potential contribution of natural sources? Or is this just a problem for Africa (and if so why?)."

According to Deng et al. (2022), the global situation from inversions for main emitters is similarly affected by the potential contribution of natural sources as well, which is difficult to estimate / separate.

We copied below Fig. 11 from Deng et al. (2022) showing that even when removing 'intact / non-managed lands' from inversions, in many countries, especially tropical countries, the inversions give a systematically much higher anthropogenic emission of $N_2O$ than inventories, suggesting that there are either missing anthropogenic sources or some 'natural' sources (e.g. conservation areas) in managed lands being underestimated by inventories.

[Figure]

Figure 11 from Deng et al. (2022).

We highlighted this point in the revised manuscript lines 896 to 902:

*"According to Deng et al. (2022), the global situation from inversions for main emitters is similarly affected by the potential contribution of natural sources as well, which is difficult to estimate and separate. Figure 11 from Deng et al. (2022) shows that even when removing "intact / non-managed lands" from inversions, in many countries, especially tropical countries, the inversions give a systematically much higher anthropogenic level of $N_2O$ than inventories, suggesting that there are either missing anthropogenic sources or some "natural" sources (e.g. conservation areas) in managed lands being underestimated by inventories."*

- "Given the importance of this uncertainty – could the paper recommend which estimate to use? Do you believe the top-down or bottom-up ones are better? At the moment it just leaves the reader to see two very different possible answers…."

Thanks for this important and complex point. Answers to this point were already indirectly included in the discussion about uncertainties in sections 3.1, 3.2 and 3.3 (lines 851 to 902), but we also added a more detailed paragraph about this specific point on section 3.6 (lines 953 to 974) as following:

*"We initially did not make any assumption regarding which approach is "better" between TD and BU method, as it actually depends on the considered gas, sector and spatial scale. Comparability between TD and BU results is not completely obvious either, as they do not represent the same processes (example of LULUCF $CO_2$ for DGVM as explained in paragraph 3.1). For $N_2O$ specifically, we highlighted in paragraph 3.3 the large uncertainty of the TD estimates, underlining the importance to separate natural $N_2O$ emissions from total estimates in order to deliver appropriate anthropogenic assessments thanks to the inversions.*

*We showed in the results of this paper that inversions in general tend to have larger uncertainties than inventories, and large differences in terms of min / max and at annual scale even among similar typologies of the methods. But at a decadal scale, they deliver reliable overall trends (with good match among the median values of various estimates on the overlapping time period) especially at the spatial scale of groups of countries and of a continent. Under such conditions, TD estimates help identify or confirm outliers / large uncertainties in inventories that may occur especially for Non-annex I countries like Africa.*

*Inversions therefore can't be a substitute but rather a complement to check trends consistency of inventories and help to identify and correct main outliers. That's why we chose BU estimates to deliver a final budget over Africa (with $CO_2$ LULUCF corrections) as synthesis figures (see Fig.12 and Fig.13 in the next paragraph). Possibilities to reduce the gap BU and TD estimates are the following: 1) For*

*inversions: to have a coarser network of surface stations and coarser spatial resolution. 2) For DGVM: see paragraph 3.1. 3) For national UNFCCC inventories: to have regularly updated activity data and use country-specific emissions data and include indirect emissions, which is not the case to date for African countries, and use expert judgment for correcting outliers as done by Grassi et al. (2022) and in this study for $CO_2$ LULUCF emissions.*"

- "Table 1: why do you list all 3 $N_2O$ inversions as different lines here, but for $CO_2$ you just have "GCB ensemble" (which is itself multiple models)?"

We suppressed the listing of all 3 $N_2O$ inversions and referred them to the $N_2O$ budget paper in Table 1 (see below lines 217-217) as visible in the marked-up revised version of the manuscript. The list of detailed products is available in the supplement document on Table S7 (below line 206).

- "Line 291 – Pongratz is a good reference here for the issues of "loss of sink capacity". https://esd.copernicus.org/articles/5/177/2014/esd-5-177-2014.pdf"

Thanks, we added the reference to this paper by Pongratz et al. (2014) and we included the quotation of her definition of "loss sink capacity" as following on lines 337 to 340:

*"Pongratz et al. (2014) delivered the following definition of "loss of sink capacity as the $CO_2$ fluxes in response to environmental changes on managed land as compared to potential natural vegetation. Historically, the potential natural vegetation would have provided a foregone sink as compared to human land use."*

We also added the reference to this author in the bibliography (lines 1337-1339):

*"Pongratz, J., Reick, C. H., Houghton, R. A., & House, J. I.: Terminology as a key uncertainty in net land use and land cover change carbon flux estimates. Earth System Dynamics, 5(1), 177-195. https://doi.org/10.5194/esd-5-177-2014, 2014."*

- "Line 339 – why do you use GWP100 numbers from AR4 (16 years old?) – can you update these to AR6?"

We used AR4 because many African countries have been following 2006 IPCC guidelines referring to AR4 GWP100 2019 refinement to IPCC guidelines which do not recommend any specific metrics, therefore we are following IPCC guidelines used by countries. We have further explained this point lines 394-398, and for information we have also added the coefficients to use to change AR4 to AR6 GWP100 values in the revised manuscript:

"*We used AR4 GWP100 because many African countries have been following the 2006 IPCC guidelines referring to AR4 GWP100 2019 refinement to IPCC guidelines, which do not recommend any specific metrics, therefore we are following IPCC guidelines used by countries. The multiplicative coefficients to use to change AR4 to AR6 GWP100 values are: 1.19 for fossil $CH_4$, 1.09 for non-fossil $CH_4$, and 0.92 for $N_2O$.*"

After our synthesis figure, Fig.13 (line 1022), we also added the following information (lines 1025-1033):

"*For information, in the supplement section Fig. S13 and Fig. S14 illustrate the differences in $MtCO_2e$ and in % for $CH_4$, $N_2O$ and for the total net GHG budget that would result from the use of AR6 GWP-100 compared to AR4 GWP-100 currently in used by UNFCCC non-Annex I countries, for the six African regions considered on Fig.13 as well as for Africa total. The net difference on the total African budget for the use of GWP-100 AR6 instead of AR4 is: +4.6%, which means a relatively small increasing impact on the net budget, with a prevailing effect of the slight increase of $CH_4$ GWP-100 in the AR6 as compared to AR4, over the strong decrease of $N_2O$ GWP-100. The two African regions that are the most impacted in terms of net budget are: Southern Countries (+7.2%) and the Horn of Africa (+6.3%). The least impacted region in terms of overall net budget with an updated AR6 GWP-100 for $CH_4$ and $N_2O$ is South Africa (+1.7%).*"

As mentioned in this paragraph, we plotted the differences between AR4 and AR6 in the supplementary Fig. S13 (see above lines 420-422) and Fig. S14 (above lines 432-434).

**REPLY TO THE ANONYMOUS REFEREE #1**

We thank Referee #1 for the valuable comments in the interactive discussion of our ESSD preprint review article and for acknowledging the "crucial" interest of this study. Below, in black color, we provide answers to the comments posted by this Referee (in blue).

"The paper provides an analysis of greenhouse gas (GHG) emissions and trends in Africa over the past three decades, focusing on evaluating different datasets and their potential for verifying official country-reported data. The study examines emissions of carbon dioxide ($CO_2$), methane ($CH_4$), and nitrous oxide ($N_2O$) using both bottom-up approaches (such as national inventories and ecosystem models) and top-down methods (including atmospheric inversions).

The findings contribute to understanding emission trends and uncertainties in Africa, which is crucial for climate policy and the goals of the Paris Agreement. Overall, the topic is interesting.

However, I have some concerns as follows:

1. In addition to providing the datasets (https://doi.org/10.5281/zenodo.7347077), this paper needs to include datasets usage (quality control method, datasets limitation, etc.)"

Thanks, we already included quality control method and datasets limitation especially in section 1 Methods and datasets (lines 179 to 401), that we recalled: *"1. Methods, and datasets and datasets usage"* (line 179) and that we completed as following:

● For BU methods :

- For PRIMAP-hist and Global Carbon Project fossil $CO_2$ emissions, we already detailed the datasets usage lines 231 to 248, respectively similar to Gütschow et al. (2021) and Friedlingstein et al. (2020).

[revised manuscript text omitted]

● For atmospheric inversions datasets and data usage:

We added lines 194 to 195 for prior fluxes data quality control: *"For preliminary data quality control, we checked the consistency of prior fluxes by plotting them separately (Fig. S1)."*

For inversion data limitation, see lines 195 to 213: *"Inversions only solve for total fluxes or at best for groups of sectors, whereas BU estimates have a larger number of sectors. In Table 2, we present the correspondence between 'sectors' defined by the TD and BU methods. For all datasets, we chose an atmospheric convention with negative values representing removals from the atmosphere (i.e. land sink). No specific standard guidelines currently exist for defining uncertainties of BU and TD data products. Given that some of our estimates are based on a small number of models / estimates, we cannot calculate the full distribution e.g. with a 95% confidence interval, but we rather reported ranges with min / max. Assuming that the unknown distributions would be Gaussian, like in Schulze et al. (2018), we could infer a 2-sigma (≈ 95%) confidence interval if we assume that min-max are equivalent to 3-sigma, but in view of the small numbers of estimates e.g. for $N_2O$ with only 3 inversions, we prefer to just give the min-max range. Moreover, for national inventories, as all African countries are non-Annex I, they do not deliver confidence intervals but Grassi et al. (2022) estimated for $CO_2$ LULUCF fluxes uncertainties of 50 % for the average of non-Annex-1 countries. Here uncertainty estimates are understood as the spread among minimum and maximum values from one methodology. A main source of uncertainty in the comparison of country-reported data with other data products is the inclusion or not of natural fluxes additionally to anthropogenic emissions sectors. For the comparability of the different data products presented in this study, we discuss only the mean value over the period of overlapping data availability. Referenced datasets are available at https://doi.org/10.5281/zenodo.7347077 (Mostefaoui et al., 2022)."*

● For quality control / data limitation on observations data used for calibration we added lines 107 to 113:

*"The African ground-based atmospheric network used by inversions is very sparse. There are only three currently active surface flasks over this whole continent, located in Namibia (Gobabeb), in the Seychelles (Mahe Island), and in South Africa (Cape Point). The one in Algeria (Assekrem) was terminated on 26/08/2020, and the one in Kenya has been inactive since 21/06/2011. The characteristics of the surface flasks in Africa, available on the NOAA website are summarized in Table S1. Inversion results are therefore uncertain due to this small number of atmospheric stations over the continent (Nickless et al., 2020)."*

The detailed surface flasks characteristics with stations, parameters, first samples date, status for the three GHG, frequency, elevation and cooperating agencies are also listed on Table S1 that we added in the supplementary section (below lines 4-5).

● For $CO_2$ inversions see lines 344 to 359:

*"We used the net land $CO_2$ fluxes excluding fossil fuel emissions (hereafter, net ecosystem exchange) from three global inversions of the Global Carbon Project that cover a long period (see Table A4 of Friedlingstein et al., 2020), including : CarbonTrackerEurope (CTRACKER-EU-v2019; van der Laan-Luijkx et al., 2017), the Copernicus Atmosphere Monitoring Service (CAMSV18-2-2019; Chevallier et al., 2005), and one variant of Jena CarboScope (JENA, sEXTocNEET_v2020; Rödenbeck et al., 2005). The GCP inversion protocol recommends to use as a fixed prior the same gridded dataset of $FCO_2$ emissions (GCP-GridFED). However, some modelers used different interpolations of this dataset, and one group used a different gridded dataset (Ciais et al., 2021). We applied a correction to the estimated total $CO_2$ flux by subtracting a common $FCO_2$ flux from each inversion (Figure S1 and Methodological Supplementary 1). The resulting land atmosphere $CO_2$ fluxes, or net ecosystem exchange, cannot be directly compared with inventories aiming to assess C stock changes, given the existence of land-atmosphere $CO_2$ fluxes caused by lateral processes. This issue was discussed by Ciais et al. (2021) and a practical correction of inversions was proposed by Deng et al. (2022) based on new datasets for $CO_2$ fluxes induced by lateral processes involving river transport, crop and wood product trade. We applied here the same correction to all $CO_2$ inversions"*

● For $CH_4$ inversions, see lines 360 to 379:

*"We used the $CH_4$ emissions from global inversions over 2000-2017 from the Global Methane Budget (Saunois et al., 2020) (Table 1). This ensemble includes 11 models using GOSAT satellite $CH_4$ total-column observations covering 2010-2017, and 11 models assimilating surface stations data (SURF) since 2000 (Table S5). Surface inversions are constrained by very few stations for Africa, while the GOSAT satellite data has a better coverage. One could thus expect GOSAT inversions to give more robust results. Inversions deliver an estimate of surface net $CH_4$ emissions, although some of them solve for fluxes in groups of sectors, called 'super-sectors'. We have not used in situ for dataset validation per se, only the GOSAT data were evaluated against Total Carbon Column Observing Network (TCCON) independent ground based total column-averaged abundance of $CH_4$ ($XCH_4$). In the inversion dataset, net $CH_4$ surface emissions were interpolated into a $0.8° \times 0.8°$ resolution, regridded from coarser resolution fluxes and separated into 'super-sectors' either using prior emission maps or posterior estimates for those inversions solving fluxes per supersector, following Saunois et al. (2020). More specifically, these five super-sectors are: 1) Fossil Fuel, 2) Agriculture and Waste, 3) Wetlands, 4) Biomass and Biofuel Burning (BBUR), and 5) Other natural emissions. We separated $CH_4$ anthropogenic emissions from inversions using Method 1 and Method 2 proposed by Deng et al. (2021). Method 1 relies on the separation calculated by each inversion except for the BBUR supersector from which wildfire emissions were subtracted based on the Global Fires Emission Dataset (GFED) version 4 (van der Werf et al., 2017). Method 2 removes from total emissions the median of natural emissions from inversions (Deng et al. 2022). The two methods gave similar results and only Method 1 was used in the results section."*

For CH$_4$, we added lines 367 to 369 to the previous version of the manuscript: *"We have not used in situ for dataset validation per se, only the GOSAT data were evaluated against Total Carbon Column Observing Network (TCCON) independent ground based total column-averaged abundance of CH$_4$ (XCH$_4$)."*

●     For N$_2$O inversions, see lines 381 to 387:

*"We used three N$_2$O atmospheric inversions from the global N$_2$O budget synthesis (Tian, 2020) and from Deng et al. (2022) ( Tables S1, S7) : PyVAR CAMS (Thomson et al., 2014), MATCM_JMASTEC (Rodgers, 2000), (Patra et al., 2018), and TOMCAT (Wilson et al., 2014; Monks et al., 2017). We used the total N$_2$O flux from inversions including natural emissions, given that natural emissions estimates are highly uncertain for Africa. Inversion results are therefore not directly comparable with the PRIMAP-hist inventory which only contains anthropogenic emissions."*

●     For metrics to compare gasses and ancillary data and data usage, see lines to 389 to 401:

*"We express emissions of non-CO$_2$ gasses in megatons of carbon dioxide equivalent (MtCO$_2$e) using the Global Warming Potential over a 100-year time horizon (GWP100) values from the fourth IPCC Assessment Reports (IPCC AR4, WGI Chapter 2, 2007), consistent with PRIMAP-hist and historical country-reported data. We used AR4 GWP100 because many African countries have been following the 2006 IPCC guidelines referring to AR4 GWP100 2019 refinement to IPCC guidelines, which do not recommend any specific metrics, therefore we are following IPCC guidelines used by countries. The multiplicative coefficients to use to change AR4 to AR6 GWP100 values are: 1.19 for fossil CH$_4$, 1.09 for non-fossil CH$_4$, and 0.92 for N$_2$O. We used population data from the United Nations population (World Population Prospects 2019, 2022), for computing per capita FCO$_2$ emissions and their disparities, based on Gini indices (Dortman et al., 1979) for measuring statistical dispersions among a given population (methodological supplementary M2). We also used African GDP data (World Bank, 2017)."*

2.     "The method for calculating trends needs to be described and the impact of different trend calculation methods on trend results needs to be discussed."

Thanks, the revised paper contains a more detailed description of the different computation methods on trends with further details.

We computed the GINI for emissions per GDP that we detailed in methodological supplementary M2 named "steps for computing the GINI index of African country emissions" page 8 of the supplement. (lines 73-101):

*"The GINI index is a metric assessing the level of dispersion and therefore the level of inequalities among the values of a given dataset. To show the inequalities of per capita emissions among the*

*African countries, we computed the continent GINI index for each of the last three decades using the Pareto principle for the following fluxes: fossil $CO_2$ per capita emissions, $CH_4$ fossil + agriculture per capita emissions, $CH_4$ from agriculture per capita emissions.*

*We computed the GINI using the Paretto method also named 20/80 or ABC method, using an excel file for the several countries' data manipulation. We obtained the GINI index (γ) thanks to the formula:*

$$Y = \frac{[(\sum_{1}^{n} y_i \times x) - 5000 ]}{5000}$$

*When γ is bigger than 0.6, it means that the area delimited by the curve of the cumulated criterion and the graph diagonal represents more than 60% of the surface of half of the graph, and that the dispersion of the dataset is high. This method was built in the 19th century based on Vilfredo Paretto's observations regarding the inequalities of repartition of the volume of housing taxes among the taxpayers (he realized that 80% of this tax was paid by 20% of the taxpayer.) The different steps that we followed to compute the GINI are detailed below:*

*1)      computation of the territorial emissions per capita in every African country,*

*2)      ranking in a decreasing order (from the highest to the smallest one),*

*3)      computation of the cumulative emissions,*

*4)      creation of a column with the cumulative emissions expressed as a percentage,*

*5)      creation of a column with a rank (integer) for those ordered emissions from the biggest to the smallest,*

*6)      conversion of this rank as a percentage in another column,*

*7)      distinction of the emissions representing less than 25% of emissions, less than 50%, and less than 75% of emissions.*

*8)      computation of the GINI index (γ) thanks to the Paretto's formula given above. »*

For estimating linear trends and their significance, we used Python to compute the correlation coefficient for median values over overlapping time periods, that we described more in detail on page 9 of the supplementary section named "Methodological supplementary M3. Computation of correlation coefficient." (lines 104-113):

*« In mathematics, the linear correlation between two variables that we can call X and Y implies that two variables have a linear relationship between each other. If there is a linear relation between two variables, it can be represented by a straight line.  To compute this linear correlation coefficient, we use the Pearson formula that is the computation of the covariance among variables (cov(X,Y)) , divided by the product of their standard deviation ($\sigma_X$ and $\sigma_Y$). Thus, we can compute the linear correlation among two variables by using the following formula:  $\rho_{(X,Y)} = cov(X,Y)/(\sigma_X \sigma_Y)$. The higher the absolute value of a linear correlation coefficient between two variables, the more the variables are linearly correlated."*

For comparing TD and BU trends, we simply used the mean values on the overlapping timeseries as mentioned in section 1 related to methods and datasets in lines 210 to 212: *"For the comparability of the different data products presented in this study, we discuss only the mean value over the period of overlapping data* availability."

3.        "As statistics play a crucial role in this study, it is important to provide further details, such as confidence intervals, to ensure its robustness."

Thanks for your comment, we agree with Referee#1 that confidence intervals are critical. Given that some of our estimates are based on a small number of models / estimates, we cannot calculate the full distribution and a 95% confidence interval (CI) but we rather reported ranges with min / max. Assuming that the unknown distributions would be Gaussian, like in Schulze et al. (2018) (see lines 199 to 213) we could infer a 2-sigma ($\approx$ 95%) CI if we assumed that min-max are equivalent to 3-sigma, but in view of the small numbers of estimates e.g. for $N_2O$ with only 3 inversions, we prefer to just give the min-max range. Moreover, for NGHGI, this is trickier and as all African countries are non-Annex I, they unfortunately do not deliver confidence intervals but Grassi et al. (2022) estimated for $CO_2$ LULUCF fluxes uncertainties of 50 % for the average of non-Annex-1 countries, which we mentioned in the text and used by default in the revised manuscript.

We extended the discussion on uncertainties started in section 1 of this paper (methods, datasets and data usage), and we added the following paragraph about the underlying data uncertainty description in the method section lines 199-213:

*"No specific standard guidelines currently exist for defining uncertainties of BU and TD data products. Given that some of our estimates are based on a small number of models / estimates, we cannot calculate the full distribution e.g. with a 95% confidence interval, but we rather reported ranges with min / max. Assuming that the unknown distributions would be Gaussian, like in Schulze et al. (2018), we could infer a 2-sigma ($\approx$ 95%) confidence interval if we assume that min-max are equivalent to 3-sigma, but in view of the small numbers of estimates e.g. for $N_2O$ with only 3 inversions, we prefer to just give the min-max range. Moreover, for national inventories, as all African countries are non-Annex I, they do not deliver confidence intervals but Grassi et al. (2022) estimated for $CO_2$ LULUCF fluxes uncertainties of 50 % for the average of non-Annex-1 countries. Here uncertainty estimates are understood as the spread among minimum and maximum values from one methodology. A main source of uncertainty in the comparison of country-reported data with other data products is the inclusion or not of natural fluxes additionally to anthropogenic emissions sectors. For the comparability of the different data products presented in this study, we discuss only the mean value over the period of overlapping data availability. Referenced datasets are available at https://doi.org/10.5281/zenodo.7347077 (Mostefaoui et al., 2022)."*

In the discussion paragraph 3.1 about uncertainties for DGVM and inversions for LULUCF $CO_2$ (lines 851-875), we also reminded how uncertainties were defined for each method while discussing "unknown-unknown' types of uncertainties.

4.    "When employing in situ surface networks for dataset validation, are there specific factors, such as latitude, longitude, climate zones, etc., that exhibit correlations with the product's quality?"

Thanks for the question. May we please ask to what line of the paper exactly does the anonymous referee #1 refer to? We have not used in situ for dataset validation per se, only the GOSAT data were evaluated against TCCON independent ground based total column $XCH_4$. See our answer to your question 1 lines 367 to 369: *"We have not used in situ for dataset validation per se, only the GOSAT data were evaluated against Total Carbon Column Observing Network (TCCON) independent ground based total column-averaged abundance of $CH_4$ ($XCH_4$)"*, and we also added lines 107 to 113: *"The African ground-based atmospheric network used by inversions is very sparse. There are only three currently active surface flasks over this whole continent, located in Namibia (Gobabeb), in the Seychelles (Mahe Island), and in South Africa (Cape Point). The one in Algeria (Assekrem) was terminated on 26/08/2020, and the one in Kenya has been inactive since 21/06/2011. The characteristics of the surface flasks in Africa, available on the NOAA website are summarized in Table S1."*

We summarized the characteristics of the surface flasks in Africa, with synthesized data from the NOAA website in the table below that we added in the supplementary section (Table S1).

| Station name, Country | Parameter | First sample date | Status for the three GHG | Frequency | Elevation (in meters above mean sea level) | Cooperating Agencies |
|---|---|---|---|---|---|---|
| Assekrem, Algeria | $CO_2$ $CH_4$ $N_2O$ | 12/09/1995 12/09/1995 12/09/1995 | Terminated since 26/08/2020 | Discrete Monthly | 2710 | Algerian National Office of Meteorology |
| Gobabeb, Namibia | $CO_2$ $CH_4$ $N_2O$ | 13/01/1997 13/01/1997 13/01/1997 | Ongoing | Discrete Monthly | 456 | Gobabeb Training and Research Center |
| Mahe Island, Seychelles | $CO_2$ $CH_4$ $N_2O$ | 15/01/1980 12/05/1983 13/06/1997 | Ongoing | Discrete Monthly | 2 | Seychelles Bureau of Standards |
| Cape Point, South Africa | $CO_2$ $CH_4$ $N_2O$ | 5/01/1980 12/05/1983 13/06/1997 | Ongoing | Discrete Monthly | 230 | South African Weather Service |

| Mt. Kenya, Kenya | CO$_2$ CH$_4$ N$_2$O | 11/02/2010 11/02/2010 11/02/2010 | Inactive since 21/06/2011 | Discrete Monthly | 3644 | Kenya Meteorological Department |
|---|---|---|---|---|---|---|

5. "Lines 89-90. BU methods and TD methods need to be further explained."

We added more detailed explanations in the revised manuscript as reproduced below (lines 92 to 107):

*"Country reports estimate GHG emissions through statistical inventories using estimates of national sectoral activity data multiplied by emissions factors, with three levels of refinements depending on countries, named Tier 1 for default emissions factors, Tier 2 for country-specific emissions factors / activity data and Tier 3 for more emissions factors / activity with tailored representation at the scale of process. Other BU inventories for assessing national emissions also exist: they are based on the same approach as country-reported inventories but use their own parameters for activity data and emissions factors coming from research groups, international statistical agencies, etc. Process-based ecosystem models developed by the research community are not used by countries. They are based on the representations of complex ecosystem processes and can also be viewed as a BU method. Besides, another approach is named "top-down" and refers to atmospheric inversions. Inversions consist in estimating causes (emissions and sinks) based on consequences (concentrations). The inverse modeling approach consists in adjusting a priori fluxes to the atmospheric transport in order to be as adjusted as possible with observation data by minimizing a cost function. This is a mathematically complex problem under constrained because every point of the globe is an unknown emission, and there is only a limited number of observations: "regularization" techniques are used to find a unique solution."*

6. "Line 124 and Line 162. Redefinition for the abbreviation "bottom-up (BU)". You have already defined it in line 89. Please recheck the manuscript to ensure that the same error does not occur."

Thanks, we made sure that acronyms / abbreviations are only defined once in the revised manuscript.